



# Diffusional growth of cloud droplets in homogeneous isotropic turbulence: DNS, scaled-up DNS, and stochastic model

Lois Thomas[1,2], Wojciech W. Grabowski[3], and Bipin Kumar[1]

[1]Indian Institute of Tropical Meteorology, Pune, India
[2]Department of Atmospheric and Space Sciences, Savitribai Phule Pune University, Pune, India
[3]National Center for Atmospheric Research, Boulder, Colorado, USA

**Correspondence:** Wojciech W. Grabowski(grabow@ucar.edu)

**Abstract.** This paper presents a novel methodology to use the Direct Numerical Simulation (DNS) to study the impact of isotropic homogeneous turbulence on the condensational growth of cloud droplets. As shown by previous DNS studies, the impact of turbulence increases with the computational domain size, that is, with the Reynolds number, because larger eddies generate higher and longer-lasting supersaturation fluctuations that affect growth of individual cloud droplets. The traditional DNS can only simulate a limited range of scales because of the excessive computational cost that comes from resolving all scales involved, that is, from large scales at which the turbulent kinetic energy (TKE) is introduced down to the Kolmogorov microscale, and from following every single droplet. The novel approach is referred to as the 'scaled-up DNS'. The scaling-up is done in two parts, first by increasing both the computational domain and the Kolmogorov microscale, and second by using super-droplets instead of real droplets. To ensure proper dissipation of TKE and scalar variance at small scales, molecular transport coefficients are appropriately scaled-up with the grid length. For the scaled-up domains, say, meters and tens of meters, one needs to follow billions of real droplets. This is not computationally feasible, and so-called super-droplets are applied in scaled-up DNS simulations. Each super-droplet represents an ensemble of identical real droplets, and the number of real droplets represented by a super-droplet is referred to as the multiplicity attribute. After simple tests showing validity of the methodology, scaled-up DNS simulations are conducted for five domains, the largest of $64^3$ m$^3$ volume using a DNS of $256^3$ grid points and various multiplicities. All simulations are carried out with vanishing mean vertical velocity and with no mean supersaturation, similarly to past DNS studies. As expected, the supersaturation fluctuations as well as the spread in droplet size distribution increase with the domain size, with the mean droplet radius variance increasing in time $t$ as $t^{1/2}$ as identified in previous DNS studies. Scaled-up simulations with different multiplicities document numerical convergence of the scaled-up solutions. Finally, we compare the scaled-up DNS results with a simple stochastic model that calculates supersaturation fluctuations based on the vertical velocity fluctuations updated using the Langevin equation. Overall, the results document similar scaling as in previous small-domain DNS simulations and support the notion that the stochastic subgrid-scale model is a valuable tool for the multi-scale simulation of droplet spectral evolution applying large-eddy simulation model.



# 1 Introduction

The impact of turbulence on the growth of cloud droplets is an important and still poorly understood aspect of cloud physics.
This is because of the wide range of spatial scales that affect droplet growth, from the Kolmogorov microscale (about a millimeter for typical atmospheric turbulence levels) to the scale of the entire cloud or cloud system. Cloud droplets grow by the diffusion of water vapor and by gravitational collision/coalescence, with the former dominating growth until droplets are large enough so the collisional growth can be initiated and eventually led to drizzle and rain formation. For the gravitational collision/coalescence, the frequency of droplet collisions depends on the droplet spectrum width. It follows that understanding
processes leading to the observed droplet spectra is important for the understanding of the rain onset. Observations of natural droplet spectra go back to the early days of aircraft cloud studies (e.g., Warner, 1969) and continue in numerous subsequent investigations (e.g., Jensen et al., 1985; Brenguier and Chaumat, 2001; Pawlowska et al., 2006; Prabha et al., 2012, among many others; see also references in Grabowski and Wang, 2013). Those observations typically show that observed droplet spectra are wider than predicted by simple models of cloud dynamics and microphysics. Reasons for such a discrepancy have
occupied cloud physics community for decades, with some of those mechanisms involving cloud turbulence.

    Srivastava (1989) was first to argue that the traditional approach to the growth of a cloud droplet population by the diffusion of water vapor, the macroscopic approach, is not appropriate, and it should be replaced by an approach where each droplet grows in response to the supersaturation in its immediate environment, the microscopic approach. Such an argument is consistent with a study by Cooper (1989) who conjectures that droplets observed at a given location within a turbulent cloud arrive
there following different trajectories and experiencing different growth histories, and that this process results in a significant broadening of the droplet spectrum. This mechanism is referred to as the eddy hopping in Grabowski and Wang (2013), see also Grabowski and Abade (2017) and Abade et al. (2018). Lasher-Trapp et al. (2005) show that eddy-hopping can indeed result in a significant spectral broadening and can simulate droplet spectra in a better agreement with in-situ observations. Cooper et al. (2013) argue that spectral broadening resulting from eddy hopping can significantly accelerate drizzle and rain formation in
cumulus clouds.

    Motivated by Srivastava (1989), Vaillancourt et al. (2001, 2002) were first to apply direct numerical simulation (DNS) approach to study diffusional growth of cloud droplets in a turbulent environment. DNS has been initially applied to turbulent particle-laden flows to study the so-called preferential concentration (or clustering) of inertial particles in turbulence (e.g., Eaton and Fessler, 1994, see also references in Shaw, 2003). Arguably, the clustering can also affect diffusional growth of
cloud droplets as argued in Shaw et al. (1998); however, see the comment on that paper by Grabowski and Vaillancourt (1999). For the DNS of cloud droplets growing by the diffusion of water vapor, the droplets respond to the supersaturation fluctuations in their immediate environment as suggested by Srivastava (1989). Limited by the computational resources, Vaillancourt et al. (2002) were only able to consider small volumes of a turbulent cloud, around 1 liter. Three sets of simulations were performed with turbulent kinetic energy dissipation rates relevant to cloud conditions and including droplet sedimentation. Earlier studies
of particle-laden turbulent flows typically exclude sedimentation (see Eaton and Fessler, 1994) but this is inappropriate for weak to moderate turbulence intensities typical for natural clouds (Grabowski and Vaillancourt, 1999). Vaillancourt et al.





(2002) simulations show a small impact on the droplet spectra: the standard deviation of the initially monodisperse droplet distribution increases very slowly with time, of the order of 0.01 $\mu$m per minute. Similar simulations reported in Lanotte et al. (2009) applying larger domains clearly show that the impact, although still relatively small (a few tenths of 1 $\mu$m), does

increase with the domain size (see Fig. 3 therein). In similar DNS simulations, Li et al. (2019) demonstrate the increase of spectral broadening with the increase of the domain size (i.e., the increasing Reynolds number) and the increase of the length of the simulations, see Figs. 3 and 4 therein. For the largest domain of $512^3$ and about a minute of the simulation time, the initially monodisperse 10 μm droplets evolve into a spectrum with about 1 μm width.

DNS simulations of Vaillancourt et al. (2002), Lanotte et al. (2009) and Li et al. (2019) are limited by the computational

domain size. As a result, simulations featuring domains larger than a fraction of a cubic meters are simply not yet possible. At the same time, as argued in Grabowski and Wang (2013) and documented in Grabowski and Abade (2017; see Fig. 4 therein) and Li et al. (2019; see Fig. 4 therein), the impact of turbulent supersaturation fluctuations on the spectral width increases with the domain size. A simple argument is that this is because the largest turbulent eddies impact most strongly the local supersaturation fluctuations and thus the spread of the droplet growth histories. From the point of view of realistic cloud

modelling, developing and validating robust subgrid-scale schemes for contemporary large eddy simulation (LES) models (i.e., featuring grid lengths of a few tens of meters) requires performing DNS-like simulations in computational domains comparable to the size of the LES grid box. Such an attempt was made in Kumar et al. (2018) who conduct DNS simulations applying up to $(2\,\text{m})^3$ domain size and report systematic analysis of the dependence of mixing processes on the DNS domain size.

To this end, we propose to use what we refer to as the "scaled-up DNS" approach. Since the largest eddies are the key

for the condensational growth, one would like to apply the DNS technique in simulations with domains much larger than currently possible. For instance, taking a $128^3$ DNS simulation with 10 cm grid length gives computational domain of $12.8^3$ cubic meters, that is, comparable to the grid volume of an LES cloud simulation. To ensure a proper dissipation of the turbulent kinetic energy (TKE) at the smallest scales, one needs to scale up the molecular viscosity with the increase of the grid length. The increased number of droplets in the large domain can be accounted for by the so-called "super-droplet" technique in which

each super-droplet represents an appropriately scaled-up number of natural droplets (referred to as the multiplicity factor, Shima et al., 2009) as already applied in the appendix of Lanotte et al. (2009) and in Li et al. (2019).

The paper is organized as follows. The next section presents the model and modelling setup. Section 3 presents a general methodology of the scaled-up DNS and discusses numerical tests of this approach. Cloud droplets are added to scaled-up DNS simulations in section 4 applying the super-droplet method. Section 5 compares DNS and scaled-up DNS supersaturation

fluctuations with those obtained from a simple stochastic model. Concluding discussion is the focus of section 6.

## 2   The model and modelling set up

The numerical code used here is that of Kumar et al. (2012, 2014). It solves evolution equations for the three velocity components ($u$, $v$, $w$), the temperature $T$ and the water vapor mixing ratio $q_v$. Cloud droplets are represented as point particles followed in space and they grow or evaporate as dictated by their local environment. Droplet collisions are not considered. The



Navier-Stokes equations are solved by a pseudo-spectral method over a cubic volume with periodic lateral boundary condi-
tions in three directions using the fast Fourier transforms. Time stepping is performed using a second order predictor-corrector
method. The code is parallelized in two dimensions and the cubic domain is decomposed into so-called pencils. The same
procedures as in Kumar et al. (2012) are followed for the initial turbulent state preparation and the turbulence maintenance.
See Kumar et al. (2012, 2014) for more details.

Two modifications have been made to the code to carry out the present study. First, we included an additional source/sink
term in the temperature equation that was missing in the original code. The term describes evolution of temperature fluctuations
affected by the vertical velocity. This effect is incorporated in the DNS through the source/sink term $-gw/C_p$, where $g$ is the
gravitational acceleration, $w$ is the local vertical velocity, and $C_p$ is the specific heat capacity of air at constant pressure. The
complete equation for the evolution of temperature fluctuations is:

$$\frac{\partial T'}{\partial t} + \mathbf{u}.\nabla T' = K\nabla^2 T' + \frac{L}{C_p}C_d - \frac{g}{C_p}w', \tag{1}$$

where $K$ is the molecular diffusion coefficient, $L$ is the latent heat of vaporization, and $C_d$ is the condensation rate. Without
the last term, the vertical velocity simulated by the DNS has no impact on the supersaturation fluctuations. Since the emphasis
in Kumar et al. (2014, 2018) was on the mixing between cloudy and clear air, this omission has a negligible impact on results
presented there. However, this term is critical for the current study.

Second, we modified the way condensation rate is calculated for a single droplet. The analytic formulation applied originally
has the form:

$$C_d = \frac{4\pi K_r \rho_w}{\rho_0\,\Delta V} S R\,\Delta t \tag{2}$$

where $S$ is the supersaturation, $R$ is the droplet radius, $\rho_0$ is the air density, $\rho_w = 10^3$ kgm$^{-3}$ is the liquid water density,
$K_r = 5.00 \times 10^{-11}$ m$^2$s$^{-1}$ is the condensational growth constant (i.e., $dR/dt = K_r S/R$ where $S$ is the supersaturation),
$\Delta V = dx \times dy \times dz$ is the grid box volume, and $\Delta t$ is the time step. To ensure mass conservation, (2) is modified to:

$$C_d\Delta t = \frac{4\pi \rho_w}{3\rho_0\,\Delta V}[R^3(t) - R^3(t - \Delta t)] \tag{3}$$

where $R(t)$ and $R(t - \Delta t)$ are droplet radii at time $t$ and $t - \Delta t$, respectively.

The coupling of the Eulerian fields and the droplets is done using trilinear interpolation. The condensation rate is calculated
for each droplet by interpolating the values of $T$ and $q_v$ from the grid points enclosing the droplet. The condensation rate
is calculated at the droplet position and then redistributed to the nearest eight grid points through a reverse procedure. The
condensation rate provides a feedback on the temperature and water vapor evolutions. Inertial effects and gravitational settling
are included in the droplet motion. More details can be found in Kumar et al. (2012).

The modeling setup follows one of the simulations discussed in Lanotte et al. (2009). We consider an initial mono-disperse
droplet distribution of 13 μm radius and the concentration of 130 cm$^{-3}$. The liquid water content (LWC) is 1.19 gm$^{-3}$. Since
the mean velocity inside the DNS domain is zero, the total cloud water does not change with time, but the initial monodisperse
droplet size distribution broadens because the supersaturation fluctuates in time and space affecting the distribution (cf. Li





et al., 2019; Saito et al., 2019). The two specific aspects are discussed in the next two section that allow extending the DNS methodology into large spatial domains.

## 3 Scaling-up DNS simulation

The intensity of turbulence is typically expressed by the turbulent kinetic energy (TKE) dissipation rate $\varepsilon$. Increasing the domain size $L$ for the same $\varepsilon$ increases kinetic energy of turbulent motions. The TKE determines velocity fluctuations and controls the supersaturation variations that play the key role in the condensational growth of cloud droplets. The TKE dependence on $\varepsilon$ and $L$ is typically expressed as (e.g., Pope, 2000):

$$E \sim (L\varepsilon)^{2/3} \tag{4}$$

In the classical DNS, the grid length has to be close to the Kolmogorov microscale $\eta$ to allow proper TKE dissipation at the smallest scales. Increasing the domain size $L$ without changing the number of grid points implies that the grid length increases as well. We will refer such simulations as "scaled-up DNS". With the increased grid length, one needs to increase molecular transport coefficients to maintain proper TKE dissipation as well as the removal of scalar fluctuations. Assuming that the domain size $L$ represents appropriate scale of energy-containing eddies, the $L/\eta$ ratio represents the flow Reynolds number

$Re$:

$$\frac{\eta}{L} \sim Re^{-3/4} \tag{5}$$

(e.g., Pope, 2000; Grabowski and Clark, 1993). Keeping the Reynolds number the same for the actual and scaled-up DNS implies that

$$\frac{L_1}{\eta_1} = \frac{L_2}{\eta_2} \tag{6}$$

The Kolmogorov microscale is given by $\eta = (\frac{\nu^3}{\varepsilon})^{1/4}$ where $\nu$ is the viscosity. Applying $\eta$ with the same TKE dissipation rate $\varepsilon$ for both $\nu_1$ and $\nu_2$ leads to

$$\frac{L_1}{\nu_1^{3/4}} = \frac{L_2}{\nu_2^{3/4}} \tag{7}$$

Eq. (7) implies that when the TKE dissipation rate is supposed to be the same in DNS and scaled-up DNS, the molecular viscosity needs to be scaled up as:

$$\nu_2 = \nu_1 \left(\frac{L_2}{L_1}\right)^{4/3} \tag{8}$$

where $\nu_1$ is the viscosity in the real DNS, $\nu_2$ is the scaled-up viscosity, $L_1$ is the real-DNS domain size, and $L_2$ is the scaled-up domain size.

We used a DNS with $256^3$ grid points to study scaling-up simulations without droplets. Real DNS was run for $L = 0.256$ m and scaled-up DNS was run with domains of sizes $L = 2.56$ m, $L = 25.6$ m and $L = 256$ m. According to (8), the viscosity





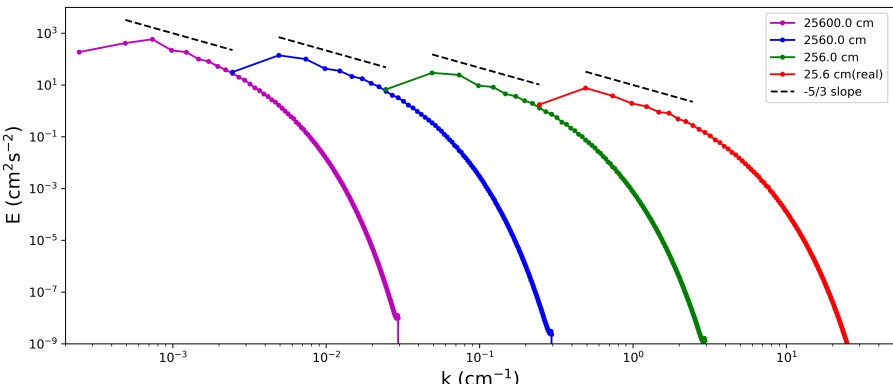

**Figure 1.** Comparison of energy spectra for real and scaled-up DNS

(taken as $\nu = 0.15$ cm$^2$s$^{-1}$ for the real DNS) has to be scaled-up by $21.54$, $464.16$ and $10{,}000$ times for $L = 2.56$ m, $L = 25.6$ m and $L = 256$ m, respectively. All simulations are forced as described in Kumar et al. (2012) applying TKE dissipation rate $\varepsilon$ of $10$ cm$^2$s$^{-3}$ as in Lanotte et al. (2009).

Fig. 1 shows energy spectra for the real $25.6$ cm DNS and the three scaled-up DNS. The black dashed lines represents the -5/3 slope expected in the inertial range. The spectral peak shifts to the left and its value increases as the domain size increases (i.e., the wavenumber $k$ decreases). The slope remains approximately similar for the four simulations.

TKE in the scaled-up simulations for the same TKE dissipation rate $\varepsilon$ should increase following the scaling originating from (4), that is,

$$E_1 = E_2 \left(L_1/L_2\right)^{2/3} \tag{9}$$

where $E_1$ and $L_1$ are for the scaled-up DNS and $E_2$ and $L_2$ are for real DNS (e.g., $L_2 = 0.256$ m and $E_2 = 20$ cm$^2$s$^{-2}$). Fig. 2 shows the evolution of TKE and TKE dissipation rate for the four simulations in Fig. 1. For the TKE evolution, dashed lines show the expected scaling based on (9). TKEs from the scaled-up DNS simulations agree with the theoretical scaled-up TKE values. To show that the DNS and scaled-up DNS feature the same TKE dissipation rate, we also show the dissipation rate calculated using the simulated enstrophy as typically done in DNS studies. The plots show that the forcing is approximately correct in the scaled-up simulations. The scaled-up simulations need to be run for longer times, with the time scale following the $L/E^{1/2}$ scaling of the large-eddy turnover time. The simulations show that the scaled-up DNS with viscosity modified according to (8) produces expected TKE.

The simulations shown in Fig. 1 and Fig. 2 feature the same dynamic range, that is, the same Reynolds number and the $L/\eta$ ratio. However, one may also consider scaled-up DNS simulations where the dynamic range is changed. For instance, one may compare simulations with the same $\varepsilon$ and $L$, and different numbers of grid points $N$ covering $L$. For such simulations, the change of the Kolmogorov microscale $\eta = L/N$ suggests the required rescaling of the dissipation coefficients. Since $\eta =$



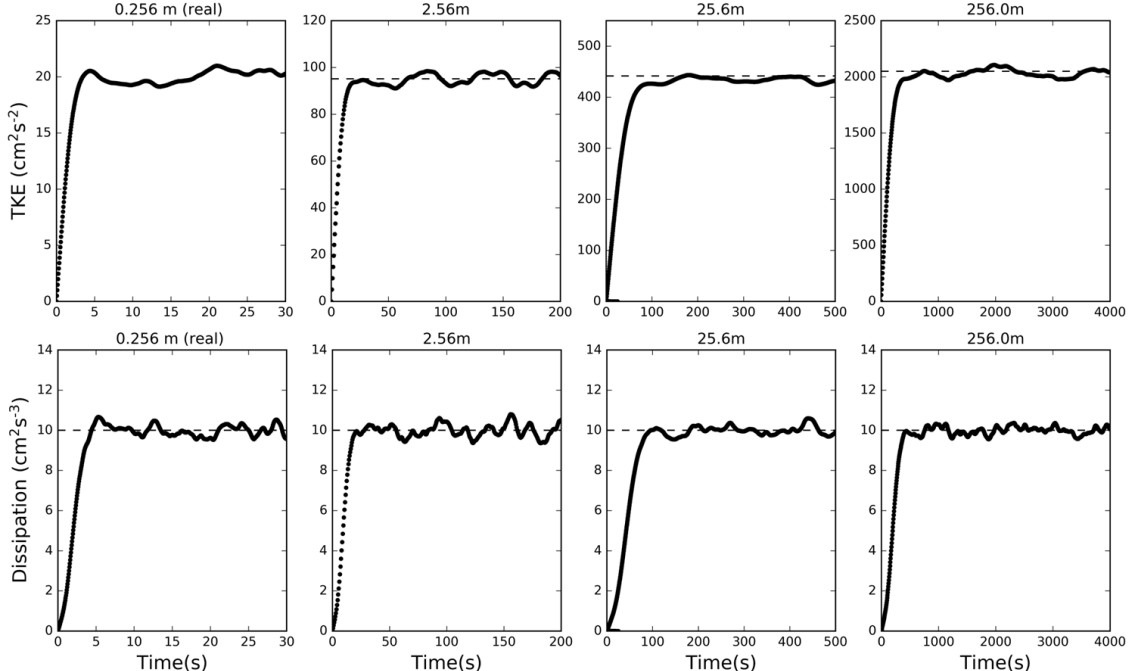

**Figure 2.** Evolution of TKE (upper panels) and TKE dissipation rate (lower panels) for four simulations mentioned in text. The dashed lines are theoritical values.

$\left(\frac{\nu^3}{\varepsilon}\right)^{1/4}$, assuming $\varepsilon$ = const gives the scaling similar to (8):

$$\nu_2 = \nu_1 \left(\frac{N_1}{N_2}\right)^{4/3} \tag{10}$$

that is, with the number of grid points rather than the domain size providing the scaling.

The scaling (10) is illustrated in Fig. 3 that shows the spectra in simulations with the domain size of either 0.512 m or

1.024 m and applying either DNS or scaled-up DNS. The spectra are obtained at final simulation times. The red lines represent spectra for the real DNS, and green and blue lines show spectra for scaled-up DNS. Scaling-up accurately predicts the energy at the largest scales, but some energy at smaller scales, still far from the dissipation, is lost. This means that the total TKE for a scaled-up DNS is slightly lower than the real DNS within the same volume. For the simulations shown in Fig. 3, TKE for $L$ of 0.512 m is 34.2, 32.0, and 26.0 $\mathrm{cm^2 s^{-2}}$ for real DNS (N=512), and scaled-up DNS with N=256 and 128, respectively. For L of

1.024 m, TKE is 55.0, 50.0, and 41.0 $\mathrm{cm^2 s^{-2}}$ for real DNS (N=1024), and scaled-up DNS with N=256 and 128, respectively. Because for the condensational growth the interest is on the largest scales as discussed in the introduction, the energy loss at smaller scales can be considered less important. However, this aspect is relevant for the comparison between scaled-up DNS and the stochastic model as discussed in section 5.



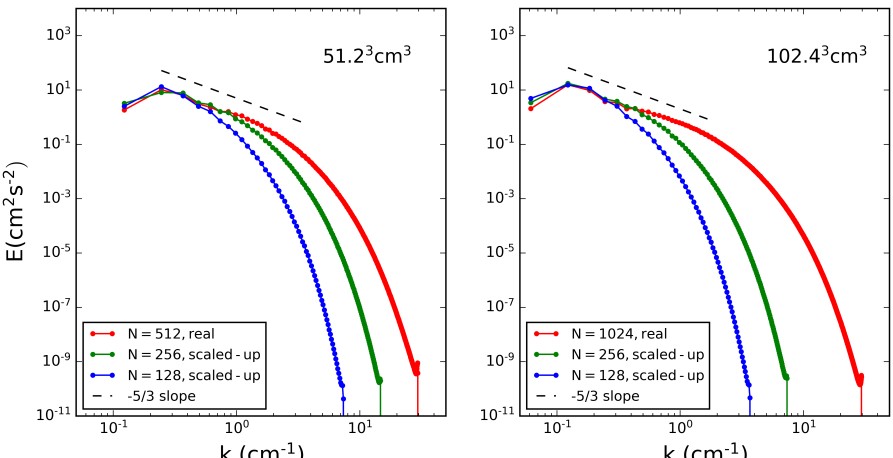

**Figure 3.** Energy spectrum comparison of real DNS and scaled-up DNS. Left/right panels correspond to $512^3/1024^3$ real DNS.

**Table 1.** Number of super-droplets and their multiplicity for real DNS domains of volume $6.4^3$ cm$^3$ and $12.8^3$ cm$^3$.

| L = 6.4 cm | L = 12.8 cm |
|---|---|
| $N_s = 34078; \mu = 1$ | $Ns = 272630; \mu = 1$ |
| $N_s = 17039; \mu = 2$ | $Ns = 54526; \mu = 5$ |
| $N_s = 6815; \mu = 5$ | $Ns = 27263; \mu = 10$ |

## 4 Applying superdroplets for the scaled-up DNS

For a scaled-up DNS, one needs to follow significantly larger number of droplets when compared to DNS. For instance, for the droplet concentration of $130$ cm$^{-3}$ one needs to follow $1.3 \times 10^{11}$ droplets for a domain of $L = 10$ m. This is not computationally feasible. To overcome this problem, one can use the so-called super-droplets (Shima et al., 2009) instead of real droplets, where each super-droplet represents an ensemble of real droplets with the same radius. Position and velocity of each super-droplet is predicted in the same way as for the real droplet. The number of real droplets represented by a single

super-droplet is referred to as the multiplicity attribute $\mu$ (Shima et al., 2009).

At the onset of simulations, super-droplets are inserted into the computational domain in the same way as regular droplets, that is, they are randomly positioned inside the domain and subsequently followed in space and time as regular droplets. The condensation rate for a super-droplet is calculated as in (2) except for an additional multiplicity factor $\mu$. The evolution of the temperature and water vapor mixing ratio fluctuations is affected by the condensation rate of super-droplets within a grid box

in the same way as regular droplets.



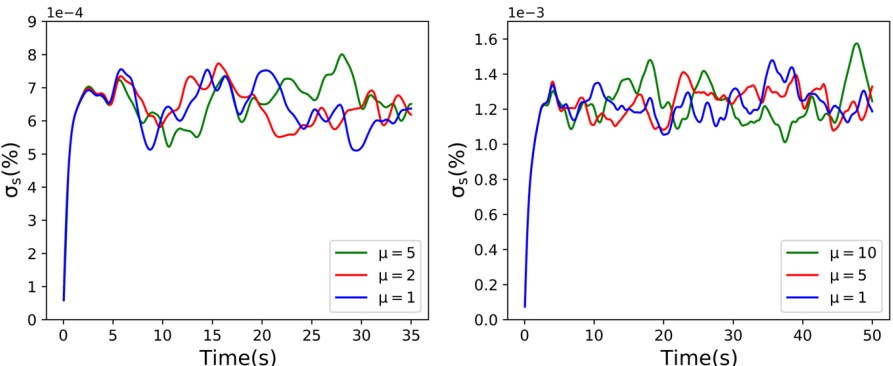

**Figure 4.** Evolution of the supersaturation fluctuation standard deviation for $6.4^3$ cm$^3$ real DNS (left panel) and $12.8^3$ cm$^3$ real DNS (right panel). Colors represent different multiplicity as marked inside each panel.

The super-droplet approach is first tested in the real DNS. Fig. 4 shows evolutions of the standard deviation of the supersaturation spatial fluctuations for real DNS of $L = 0.064$ m ($64^3$ grid points) and $L = 0.128$ m ($128^3$ grid points) with different multiplicity parameter. Number of superdroplets used and their multiplicity are listed in Table 1. For randomly distributed real droplets, the two simulations have droplets in one out of about eight grid boxes. For super-droplets in $64^3$ simulations, this

number changes to one out of about 15 grid boxes for $\mu = 2$ and about 38 grid boxes for $\mu = 5$. For the $128^3$ simulations, super-droplets are on average in one out of about 43 and 87 grid boxes for $\mu = 5$ and 10, respectively. The mean supersaturation is close to zero as expected (not shown). Supersaturation standard deviations fluctuate similarly in all simulations with the mean values close among all multiplicities. The mean value of the standard deviation is larger for the larger domain in agreement with simulations discussed in Lanotte et al. (2009).

In general, the multiplicity value should be decided carefully because too large multiplicity results in too many grid boxes without droplets when compared to real droplets and this may cause undesirable effects in the mean supersaturation and its spatial variability. In the two DNS cases, slight deviations in the mean supersaturation are present, although the simulations are not long enough to document the impact with confidence. For the scaled-up DNS, the number of droplets is in billions and we have to select higher multiplicity values to make computations feasible. The evolution of the radius squared ($R^2$) standard

deviation ($\sigma_{R^2}$) from the above simulations with droplets and super-droplets is shown in Fig. 5. Initially (i.e., at t = 0), the distributions are monodisperse (i.e., $\sigma_{R^2} = 0$). Supersaturation fluctuations in response to local vertical velocity fluctuations lead to the increase of $\sigma_{R^2}$ in time. After some initial adjustment, the increase approximately follows the $t^{1/2}$ scaling with $t$ being the time from the start of the simulation. This agrees with the study by Sardina et al. (2015) who applied a stochastic model and DNS. Similar result is also shown in Li et al. (2019) and Saito et al. (2019). As expected, the $\sigma_{R^2}$ values are larger

for the larger domain, in agreement with Fig. 3 in Lanotte et al. (2009) and Li et al. (2019).

After the super-droplet technique is tested in DNS, the same method is used in scaled-up DNS. In general, one may expect that if the multiplicity is increased beyond a certain value, the results will start deviate from those with a low multiplicity





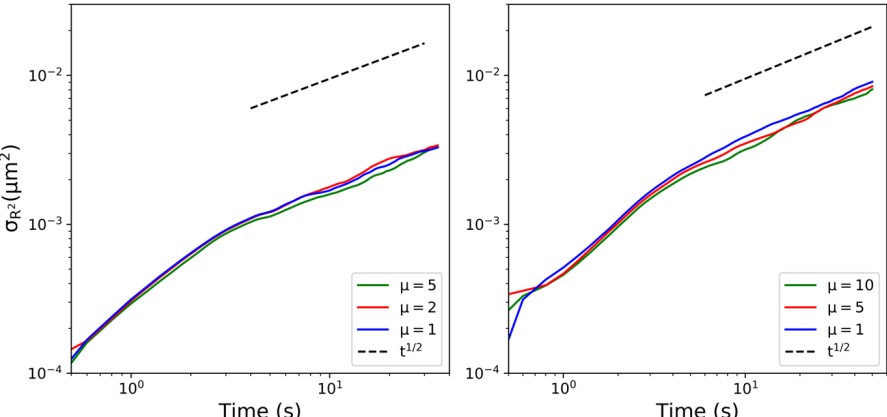

**Figure 5.** Evolutions of the radius squared standard deviation ($\sigma_{R^2}$) for real DNS with different multiplicity parameters. Left/right panels corresponds to $64^3/128^3$ real DNS ($L = 6.4/12.8$ cm). Colors represent different multiplicity as marked inside each panel.

featuring a larger number of super-droplets. However, high multiplicity is desirable to reduce the number of super-droplets that need to be followed. For the DNS, the number of super-droplets was shown to be relatively low to maintain similarity between

real droplet and super-droplet solutions (see Fig. 4 and Fig. 5; as low as one super-droplet in a few dozen of grid volumes). With scaled-up DNS, one might expect a different requirement because of a stronger local forcing of the supersaturation due to higher TKE and thus larger vertical velocities.

For the scaled-up DNS study, we apply a $256^3$ domain to represent volumes with characteristic lengths of several meters. The TKE is scaled as explained in section 3 with relevant parameters listed in table 2. As the table shows, scaled-up DNS

simulations typically have a relatively small number of super-droplets per grid box, similarly to DNS. This is because of computational efficiency considerations. However, one may question such an approach because scaled-up DNS include a large number of real droplet (e.g., $\sim 10^9$ for scaled-up DNS with 1 cm grid length and much larger numbers for scaled-up DNS with larger grid lengths). To show that the standard deviation of the supersaturation spatial distribution is not affected by the small number of super-droplets considered in the scaled-up DNS simulations, we included additional simulations (shown in bold in

the Table 2) that include about 10 super-droplets per grid volume and follow about 160 millions of super-droplets. Although arguably still a small number, 10 super-droplets per grid box is the number considered in one of the sensitivity simulations in Grabowski (2020). By comparing results of simulations with various numbers of super-droplet per grid volume, Grabowski (2020) shows that the number as small as 10 is sufficient to reasonably represent condensational growth of natural droplets in idealized simulations of laboratory chamber experiments. Fig. 6 and Fig. 7 present evolutions of the mean supersaturation and

standard deviation of its spatial distribution for the scaled-up simulations from Table 2. The five scaled-up domains shown in the table and figures correspond to the domain size $L$ of 2.56, 6.4, 12.8, 25.6 and 64 m. Note that the simulations extend to times of several minutes, that is, a significant fraction of a small convective cloud life cycle. All simulations except 12.8 and 64 m are run with two different multiplicities for super-droplets as listed in Table 2. As Fig. 6 shows, the mean supersaturation





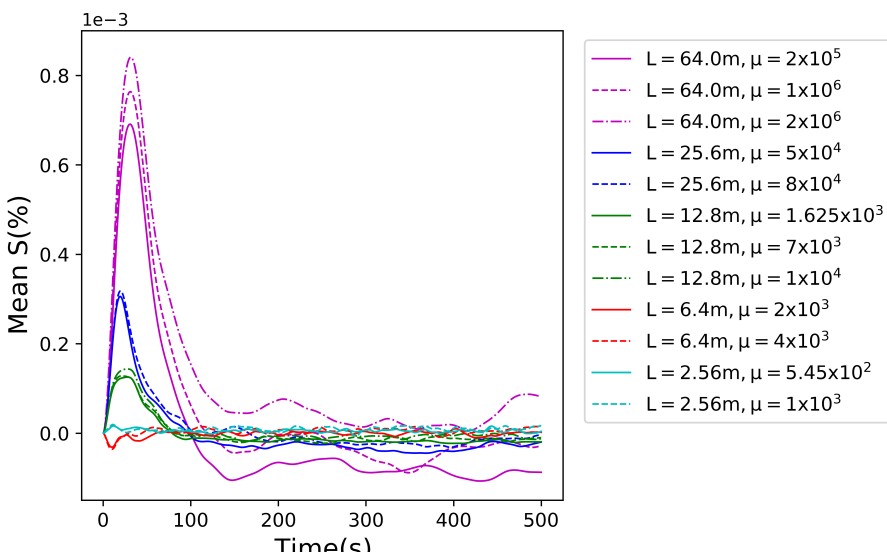

**Figure 6.** Evolution of the mean supersaturation for various scaled-up domains. Colors represent different domain size; different line styles correspond to different multiplicities. The additional simulation of 10 super-droplets per grid volume is only shown for $12.8^3$ m$^3$ and $64.0^3$ m$^3$ volumes.

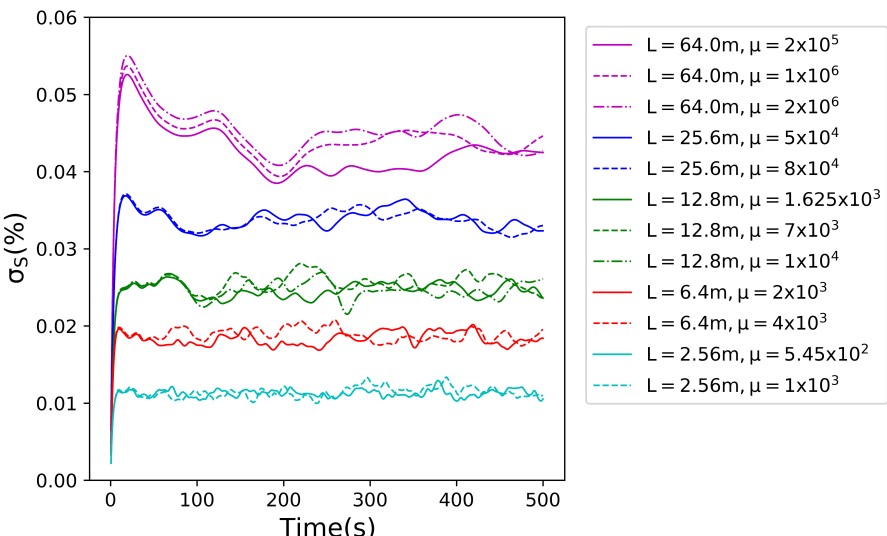

**Figure 7.** Evolution of standard deviation of supersaturation fluctuations for different domain sizes. Colors represent different domain size; different line styles correspond to different multiplicities. The additional simulation of 10 super-droplets per grid volume is only shown for $12.8^3$ m$^3$ and $64.0^3$ m$^3$ volumes.





**Table 2.** Details of DNS and scaled-up DNS. From left to right: domain length $L$, grid length $l$, viscosity $\nu$, Turbulent kinetic energy $E$, number of superdroplets in the domain $N_s$, multiplicity $\mu$, number of superdroplets per grid volume $N_s/N^3$

|  | $L$(cm) | $l$(cm) | $\nu(\mathrm{cm}^2\mathrm{s}^{-1})$ | $E(\mathrm{cm}^2\mathrm{s}^{-2})$ | $N_s$ | $\mu$ | $N_s/N^3$ |
|---|---|---|---|---|---|---|---|
| Real DNS | 25.6 | 0.1 | 0.15 | 20.0 | $\sim 2.2 \times 10^6$ | 1.0 | 0.13 |
| Scaled-up DNS | 256.0 | 1.0 | 3.231 | 94.0 | $\sim \mathbf{1.7 \times 10^8}$ | **13.0** | **10.0** |
|  |  |  |  |  | $\sim 4.0 \times 10^6$ | $5.45 \times 10^2$ | 0.24 |
|  |  |  |  |  | $\sim 2.2 \times 10^6$ | $1.0 \times 10^3$ | 0.13 |
|  | 640.0 | 2.5 | 10.965 | 171.0 | $\sim \mathbf{1.7 \times 10^8}$ | **203.125** | **10.0** |
|  |  |  |  |  | $\sim 1.7 \times 10^7$ | $2.0 \times 10^3$ | 1.01 |
|  |  |  |  |  | $\sim 8.5 \times 10^6$ | $4.0 \times 10^3$ | 0.51 |
|  | 1280.0 | 5.0 | 27.630 | 270.0 | $\sim \mathbf{1.7 \times 10^8}$ | $\mathbf{1.63 \times 10^3}$ | **10.0** |
|  |  |  |  |  | $\sim 3.9 \times 10^7$ | $7.0 \times 10^3$ | 2.32 |
|  |  |  |  |  | $\sim 2.7 \times 10^7$ | $1.0 \times 10^4$ | 1.61 |
|  | 2560.0 | 10.0 | 69.624 | 420.0 | $\sim \mathbf{1.7 \times 10^8}$ | $\mathbf{1.3 \times 10^4}$ | **10.0** |
|  |  |  |  |  | $\sim 4.4 \times 10^7$ | $5.0 \times 10^4$ | 2.62 |
|  |  |  |  |  | $\sim 2.7 \times 10^7$ | $8.0 \times 10^4$ | 1.61 |
|  | 6400.0 | 25.0 | 236.235 | 750.0 | $\sim \mathbf{1.7 \times 10^8}$ | $\mathbf{2.03 \times 10^5}$ | **10.0** |
|  |  |  |  |  | $\sim 3.4 \times 10^7$ | $1.0 \times 10^6$ | 2.03 |
|  |  |  |  |  | $\sim 1.7 \times 10^7$ | $2.0 \times 10^6$ | 1.01 |

for all five scaled-up cases is close to zero after the initial spike. The spike magnitude increases as the domain size increases,

and it is slightly larger for the higher multiplicity. Higher multiplicity also causes larger fluctuations after the initial spike, but the mean does not seem to be significantly affected. The standard deviation shown in Fig. 7 increases with the domain size as expected. For all domains, standard deviations are similar for various multiplicities. In particular, based on 12.8 and 64 m simulations, the low number of super-droplets per grid volume (desirable for computational efficiency) seem to insignificantly impact the supersaturation statistics.

To further study the impact of the multiplicity, additional scaled-up DNS simulations are run with $256^3$ grid points for a domain of size $12.8^3$ m$^3$. All simulations are listed in Table 3, with some already considered in Table 2 and Fig. 6 and Fig. 7. Total number of real droplets for the $12.8^3$ m$^3$ domain with droplet concentration of $130$ cm$^{-3}$ is about $2.7 \times 10^{11}$. The grid volume of the $256^3$ grid points and $12.83$ m$^3$ simulation is $125$ cm$^3$. When multiplicity is 1625, the number of super-droplets



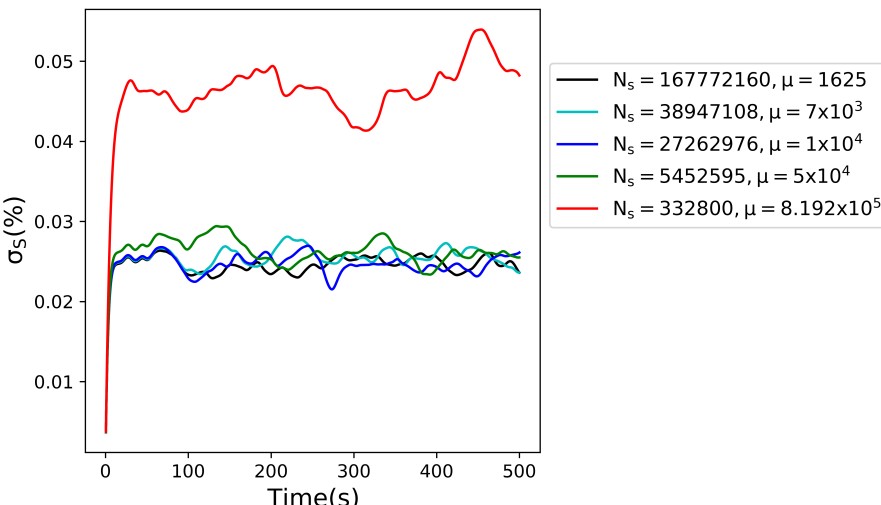

**Figure 8.** Standard deviation of supersaturation fluctuations for $12.8^3$ m$^3$ scaled up domain. Colors indicate different multiplicities.

is close to 170 millions and there are on average 10 super-droplets per grid volume. When $\mu$ is $7 \times 10^3$, the number of super-

droplets is close to 40 millions and there are on average about 2.3 super-droplets per grid volume. When $\mu$ is further increased to $1 \times 10^4$, the number of super-droplets per grid volume decreases to about 1.6. For $\mu = 5 \times 10^4$, the number further decreases to about 0.32 (i.e., a super-droplet in about 3 grid volumes). Finally, for $\mu = 8.192 \times 10^5$, a super-droplet is approximately in one out of 50 grid volumes.

Results obtained from these simulations are shown in Fig. 8 with some results already shown in Fig. 7. As the figure shows,

only the largest multiplicity, with a super-droplet in one out of 50 grid volumes differs significantly from other simulations. The highest multiplicity simulation also results in the non-zero mean supersaturation (not shown). Note that for real DNS (Table 2 and Fig. 4), having a droplet in one of several dozens of grid volumes still results in supersaturation fluctuations in agreement with real droplets. This suggests that the maximum multiplicity that can be used in scaled-up DNS depends on the domain size. This perhaps should not be surprising because the magnitude of the vertical velocity perturbation and thus the supersaturation

forcing increases with the domain size. Results for the largest domain considered in the current study ($64^3$ m$^3$) suggest that the multiplicities selected for the scaled-up DNS provide robust (i.e., independent of the multiplicity) outcomes.

As shown in Fig. 9, evolutions of the radius squared standard deviation $\sigma_{R^2}$ for scaled-up DNS domains follows the same trend as in the real DNS, that is, the standard deviations increase in time $t$ as $t^{1/2}$. The results are shown for the five scaled-up cases mentioned above. Scaled-up DNS for each domain was run for three different multiplicity values, one of them being

10 super-droplets per grid volume. The error bars correspond to the standard deviation among realizations with different multiplicities. Overall, the scatter resulting from different multiplicities is relatively small. The key result in Fig. 9 is that the spectral width increases with the domain size. For domain sizes of a few tens of meters, the spectral width after a few minutes





**Table 3.** Number of super-droplets and multiplicity for different $12.8^3$ m$^3$ scaled-up domain simulations.

| Number of superdroplets($N_s$) | Multiplicity ($\mu$) |
| --- | --- |
| $\sim 1.6 \times 10^7$ | 1625 |
| $\sim 3.9 \times 10^7$ | $\sim 7 \times 10^3$ |
| $\sim 2.7 \times 10^7$ | $\sim 1 \times 10^4$ |
| $\sim 5.4 \times 10^6$ | $\sim 5 \times 10^4$ |
| $\sim 3.3 \times 10^5$ | $\sim 8 \times 10^5$ |

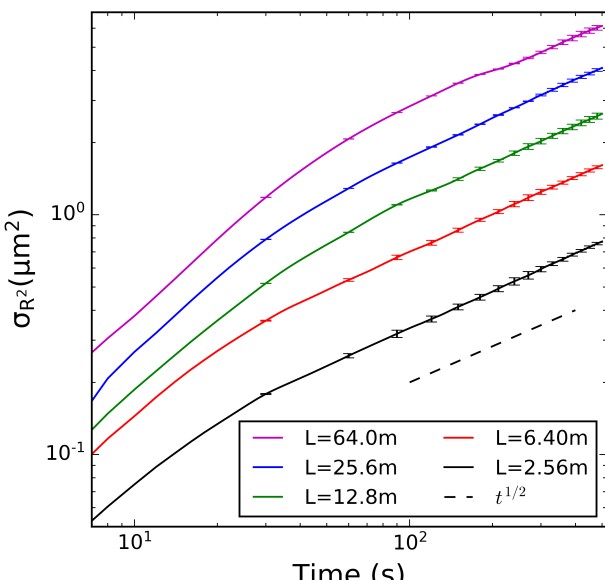

**Figure 9.** Evolutions of the radius squared standard deviation $\sigma_{R^2}$ for different domain sizes in the scaled-up DNS simulations. Horizontal bars along each line show variability resulting from different multiplicity used for each domain size.

reaches values of 1-2 μm that is comparable to those observed in near-adiabatic cores of small cumuli (e.g., Jensen et al., 1985) or subtropical stratocumulus (e.g., Pawlowska et al., 2006).

## 5   Stochastic model

We apply the stochastic model similar to that in Grabowski and Abade (2017) to simulate fluctuating supersaturation and compare results to the real and scaled-up DNS. The fluctuating in space supersaturation in the dynamic simulations (i.e., real DNS or scaled-up DNS) is modelled as independent realizations of the fluctuating in time supersaturation. For each realization, the supersaturation fluctuations are driven by the vertical velocity fluctuations as given by the Ornstein-Uhlenbeck process (e.g.,



Pope, 1994). In its finite difference implementation, the velocity perturbations are updated as in Grabowski and Abade (2017):

$$w'(t + \delta t) = w'(t)e^{-\delta t/\tau} + \sqrt{1 - e^{-\frac{2\delta t}{\tau}}}\, \sigma_{w'}\psi \tag{11}$$

where $\delta t$ is the model timestep, $\sigma_{w'}^2$ is the vertical velocity variance obtained from TKE as

$$\sigma_{w'}^2 = \frac{2}{3}E \tag{12}$$

$\psi$ is a Gaussian random number with zero mean and unit variance generated at every time step, $\tau$ is the eddy turnover time calculated as

$$\tau = \frac{L}{(2\pi)^{1/3}}\left(\frac{C_\tau}{E}\right)^{1/2} \tag{13}$$

where $C_\tau$ is a constant equal to 1.5 as in Lasher-Trapp et al. (2005). Supersaturation fluctuations evolve according to the equation

$$\frac{dS'}{dt} = a_1 w' - \frac{S'}{\tau_{relax}} \tag{14}$$

where $w'$ is the vertical velocity perturbation evolving as in (11), $a_1$ is a temperature-dependent numerical coefficient and $\tau_{relax}$ is the phase relaxation time that depends on the temperature and pressure as well as on the droplet concentration and mean radius. For the conditions considered in this study, $a_1 = 4.753 \times 10^{-4}$ m$^{-1}$ and $\tau_{relax} = 3.513$ s.

The stochastic model used here applies 1000 realizations, each starting from a random velocity perturbation [i.e., $\sigma_{w'}\psi$ as in (11)] and zero supersaturation, and run for 6 eddy turnover times. The time step in (14) is taken as one thousandth of the eddy turnover time. The number of realizations is sufficient to give results that change insignificantly when the number is further increased. Subsequently, the standard deviation of the supersaturation temporal evolution for each realization is derived. Its mean value averaged over all realizations together with the standard deviation among realizations is used in the comparison with the DNS and scaled-up DNS simulations. Fig. 10 shows the standard deviation of supersaturation fluctuations ($\sigma_S$) derived from the stochastic model as explained above for different domain sizes together with similar results from the DNS and scaled-up DNS dynamic simulations. The first five points ($L = 0.064, 0.128, 0.256, 0.512, 1.024$ m) correspond to the real DNS, whereas the last five points ($L = 2.56, 6.4, 12.8, 25.6, 64.0$ m) correspond to scaled-up DNS. The stochastic model uses TKE simulated by either DNS or scaled-up DNS, and for $L$=0.512 and 1.02 4m by both. The vertical lines for the stochastic model show twice the standard deviation among the realizations. The red circles in the left part of the figure are from DNS simulations. Standard deviations from different multiplicities as shown in Fig. 4 are smaller than the circle radius and thus they are not shown. The blue symbols are for scaled-up $256^3$ DNS simulations; standard deviations from different multiplicities are again smaller than the radius. Finally, the red circles for $L = 64$ m show $\sigma_S$ in scaled-up DNS simulations with grids of $128^3$ and $512^3$.

Overall, the stochastic model seems to reasonably represent the scale dependence of the supersaturation fluctuations. At small scales (i.e., $L = 0.064$ and $0.128$ m), DNS seems to underestimate supersaturation fluctuations. Arguably, this is because





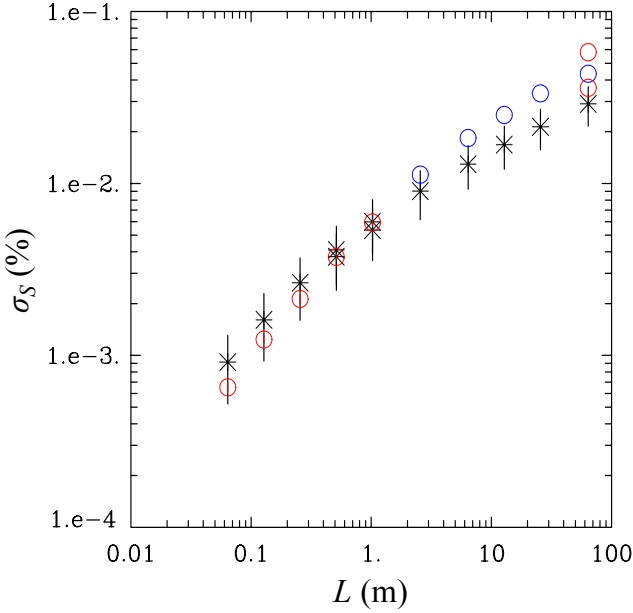

**Figure 10.** Standard deviation of supersaturation fluctuations in DNS and scaled-up DNS (colour circles) and the stochastic model (black stars). Vertical lines for the stochastic model represent variability among individual realizations. For DNS and scaled-up DNS, the variability comes from different multiplicity for super-droplets; it is not shown as it is smaller than the symbol size. Red circles in the left half are for DNS with three different multiplicities as in Fig. 4. Blue circles are for $256^3$ scaled-up DNS simulations. Red circles for $L = 64$ m are for $128^3$ (lower symbol) and $512^3$ (upper symbol) scaled-up DNS simulations. The two data points for the stochastic model with $L = 0.512$ m and $L = 1.024$ m come from applying TKE from either DNS or scaled-up DNS.

of the small Reynolds number and thus a poor separation between forcing and dissipation scales. For scaled-up DNS, the stochastic model underestimates supersaturation fluctuations and the spread between the scaled-up DNS and stochastic model increases with the increase of the spatial scale.

There are a few reasons for the discrepancy between the stochastic model and scaled-up DNS. First, stochastic model uses
TKE obtained from the scaled-up DNS. However, scaled-up DNS features reduced TKE when compared to the real DNS as documented in section 3. Allowing more TKE on input for the stochastic model would shift the stochastic model results upwards, that is, closer to the scaled-up DNS. But increasing the Reynolds number in the scaled-up DNS increases $\sigma_S$ as well. This is illustrated by three data points for $L = 64$m scaled-up DNS with $128^3$, $256^3$, and $512^3$ simulations. Second, scaled-up DNS excludes scales of motion that are smaller than the scaled-up Kolmogorov microscale. For instance, for $L =$
25.6m and $256^3$ simulation, the scaled-up Kolmogorov microscale is 0.1 m. Hence, scales of motion between 10 cm and 1 mm are excluded when compared to the real DNS. Arguably, these small-scale motions in real DNS can affect supersaturation fluctuations and reduce $\sigma_S$. Such an argument seems to be contradicted by the results with $L = 64$m because $\sigma_S$ increases, not decreases, between $128^3$, $256^3$, and $512^3$ scaled-up DNS.. However, it is unclear if the increase of $\sigma_S$ with the further increase of the Reynolds number (i.e., the number of grid points) continues once real DNS is approached with further increase of the





simulation towards the $64000^3$ real DNS limit. Finally, one might argue that assuming a Gaussian vertical velocity distribution in (11) is an increasingly poor assumption with the increase of the domain size. Higher frequency of large vertical velocity perturbations (i.e., above the Gaussian distribution) should result in larger supersaturation fluctuations.

## 6    Discussion and Conclusions

This study presents a novel modelling methodology that extends the traditional technique to simulate homogeneous isotropic

turbulence, the direct numerical simulation (DNS). DNS is typically used for small-scale simulations applying grid lengths of the order of the Kolmogorov microscale, that is, about a millimeter for typical levels of atmospheric turbulence. Such a choice allows proper dissipation of the turbulent kinetic energy (TKE) that cascades through the inertial range from large scales where TKE is introduced. To reach domain sizes of about 1 cubic meter and beyond with a grid length of about 1 mm requires tremendous computation resources, with simulations featuring spatial scales of tens of meters and beyond (i.e., volumes of

1,000s of cubic meters and larger) impossible for a foreseeable future. At the same time, one should expect that the largest turbulent eddies affect the diffusional growth of cloud droplets most significantly because such eddies feature the largest and the longest-lasting vertical velocity and supersaturation perturbations. It is thus desirable to have a modelling approach similar to the traditional DNS, but capable of reaching significantly larger spatial scales, say, tens and hundreds of meters.

This paper presents such an approach. The key idea is simple: rather than assuming that the dynamic model grid length is the

Kolmogorov microscale $\eta$, we start with the DNS domain size L and adjust the Kolmogorov microscale given the computational resources. For instance, for L = 100 m and $512^3$ simulation, $\eta \simeq 0.2$ m. However, to use a traditional DNS code one needs to allow proper TKE dissipation (as well as the scalar variance removal) at the smallest scales. It follows that the molecular transport coefficients from the traditional DNS need to be properly increased. The Reynolds number similarity is applied to develop a proper scaling, see (8) and (10). The modified modelling approach is referred to as the scaled-up DNS. Section 2

presents numerical simulations applying the spectral DNS code that document robustness of the scaled-up DNS technique. We show that DNS and scaled-up DNS simulations with the same Reynolds number (i.e., the same $L/\eta$ ratio) forced to maintain the same TKE dissipation rate feature the expected TKE scaling (4). However, when real DNSs are replaced by scaled-up DNSs with a reduced Reynolds number (i.e., keeping L the same and increasing $\eta$), a small fraction of the TKE is lost. As one might expect, the closer scaled-up DNS $L/\eta$ ratio is to the real DNS ratio, the closer are TKEs between the two simulations

(cf. Fig. 3 and its discussion).

For simulations targeting growth of cloud droplets in homogeneous isotropic turbulence, the scaled-up DNS faces the problem of a large number of droplets that need to be followed inside the computational domain. For instance, a cube volume with $L = 100$ m and droplet concentration of $100 \, \text{cm}^{-3}$ contains about $10^{14}$ droplets. Following all of them is computationally not possible. We apply a method already used in Lanotte et al. (2009) and in Li et al. (2019) and referred to as the super-droplet

method in Shima et al. (2009). A super-droplet represents an ensemble of real droplets with the same radius; position and velocity of each super-droplet is predicted in the same way as for the real droplet. The number of real droplets represented by a single super-droplet is referred to as the multiplicity attribute (Shima et al., 2009). The multiplicity attribute is included in the





condensation rate calculations. The super-droplet method was first tested in real DNS and then implemented in the scaled-up DNS. Real DNS with $L = 0.064$ m ($64^3$ grid points) and $L = 0.128$ m ($128^3$ grid points) and with different multiplicity param-

eters give consistent results even if the multiplicity parameter results in a super-droplet present in one out of a few dozen grid boxes. For scaled-up DNS (and likely for the real DNS as well), there is an upper limit for the multiplicity parameter before supersaturation fluctuations start deviating from the value obtained with lower multiplicities. Scaled-up DNSs presented here suggest that there should be at least a few super-droplets per grid box for approximately converged solutions. Such an estimate agrees with the result of idealized laboratory cloud chamber simulations reported in section 4 of Grabowski (2020).

The scaled-up DNSs starting from unimodal droplet distribution with no mean ascent (i.e., as in Lanotte et al., 2009; Li et al., 2019) extend the validity of the scaling relationship obtained previously in either DNS simulations (e.g., Li et al., 2019; Saito et al., 2019) or in stochastic model simulations (Sardina et al., 2015). The scaling implies that the standard deviation of the droplet radius squared increases in time $t$ as $t^{1/2}$. DNS results of Li et al. (2019) show that the evolution of the droplet distribution spread depends on the Reynolds number (i.e., the DNS domain) and is insensitive to the TKE dissipation rate. The

Reynolds number dependence is consistent with the eddy hopping argument and the dominating impact of the largest eddies for the spread of the droplet size distribution. The standard deviation of the droplet radius squared increases in our simulations as $t^{1/2}$ as well, with systematically larger values for larger scaled-up DNS domains as shown in Fig. 9.

Finally, we also consider supersaturation fluctuations in a simple stochastic model of a droplet ensemble (Grabowski and Abade, 2017) and compare the fluctuations to those simulated in DNS and scaled-up DNS. The key advantage of the stochastic

model is that its computational cost is just a tiny fraction of a DNS simulation. The simulation time of the stochastic model is typically a mere few seconds on a laptop computer comparing to hours of wall clock time of high performance massively parallel computer applied in DNS and scaled-up DNS simulations. As argued in Grabowski and Abade (2017), the stochastic model provides a simple and physically appealing approach to multiscale large-eddy simulation of a cloud applying Lagrangian particle-based microphysics (see Grabowski et al., 2019, for a discussion of the Lagrangian microphysics).

The scaled-up DNS methodology presented here was developed with diffusional growth of cloud droplets in mind. The next step can be to apply this approach in a rising parcel simulations as in Grabowski and Abade (2017) to understand the impact of turbulence on the cloud condensation nuclei activation/de-activation near the cloud base (see discussion in Abade et al., 2018). One can argue that scale-dependent supersaturation fluctuations can induce significant droplet concentration heterogeneities at the cloud base that arguably affect droplet growth aloft. One may also consider applying the scaled-up DNS to the problem of

droplet collisions. However, since collisions between cloud droplets take place at sub-Kolmogorov scales, applying scaled-up DNS for turbulent collisions is not straightforward. Finally, one can also consider applying scaled-up DNS in simulations of the turbulent entrainment and mixing similar to those discussed in Kumar et al. (2018). Such simulations would extend the still relatively small-domain DNS simulations into domain sizes comparable to the large entraining eddies in natural cumuli as discussed in Grabowski and Clark (1993). We hope to explore some of these research directions in the future.



*Data availability.* Data supporting the study is available at https://www.tropmet.res.in/~majfiles/Lois-Thomas/ACPfiles.zip and can be accessed upon request.

*Author contributions.* LT ran simulations and performed data analysis under the supervision of GW and BK. GW and LT developed the idea of scaled-up DNS. All three authors were involved in preparing the manuscript. BK helped in accessing and using HPC system of IITM that was used to run the DNS.

*Competing interests.* The authors declare that they have no conflict of interest.

*Acknowledgements.* DNS and scaled-up DNS were performed on HPC system Aditya at the Indian Institute of Tropical Meteorology, Pune, India. BK acknowledges support from the Ministry of Earth Science, Government of India, for providing HPC facilities to conduct the required simulations for this work. LT acknowledges NCAR's Advanced Study Program and Mesoscale and Microscale Meteorology Laboratory support for her 6-month visit to NCAR during which most of the research described here was completed. WWG acknowledge
partial support from the U.S. DOE ASR Grant DE-SC0020118. NCAR is sponsored by the National Science Foundation.





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
