# Peer review of "Diffusional growth of cloud droplets in homogeneous isotropic turbulence: DNS, scaled-up DNS, and stochastic model"

_Atmospheric Chemistry and Physics, 2020_

## Referee Comment (RC1) · Anonymous Referee #1 · 9 Apr 2020

Comments to the manuscript with ID number "acp-2020-159"

General comments:

My understanding of the "scaled-DNS" presented in this manuscript is that one can simulate a large domain size by artificially increasing the kinetic viscosity of the airflow. The Reynolds number is kept unchanged in such simulations so that the computational cost is still feasible. Therefore, one can study how the supersaturation fluctuations can be affected by the large eddies. This is plausible as the small scales do not matter for the supersaturation fluctuations. The authors further tested the application of the superdroplet approach in such a setup to tackle the condensation process.

I would recommend the publication of this manuscript after the authors carefully discuss the following comments.

The author addressed both in the abstract and in the conclusion that this is a "novel methodology". This statement should be treated carefully for the following reasons: 1. The method presented in this manuscript is DNS with large artificial kinetic viscosity, which is not a new method. 2. Mellado et al also used the same treatment (section 3 of Mellado et al).

Specific comments:

1. The paper by Mellado et al applied the same idea. Can the authors compare their work with the one of Mellado et al? 2. Can the authors check Eq.3 again? If $dR/dt=KrS/R$, then $R^2/2=KrS$, together with Eq.2, you will get a pre-factor of 2 instead of 4/3 in Eq.3, right? 3. L.125: Should the intensity of turbulence determined by the single parameter, Reynolds number? The energy dissipate rate is a small-scale quantity which describes how fast energy dissipates in the dissipation range of turbulence. In other words, it characterizes how vigorous the small eddies of turbulence are. It is calculated from the trace of the strain tensor. This aspect is also discussed by the authors in the paragraph just below Eq.10. Did you mean the energy transfer rate here, which is the rate energy transfers from large to small eddies in 3-D turbulence? 4. Can the authors normalize the energy spectrum in the same way as the one of Fig.1 of Li et al. (2019)? If the Reynolds number is the same, the normalized spectrum should collapse on top of each other; 5. Why is there an initial spike in Fig.6? I don't understand why it is different for different domain sizes. The water mass loading is the same for all the simulations, right? 6. As the integral time scale is different for simulations with different domain size, could it be an idea that the authors normalize the time axis by the integral time scale?

Technical corrections: 1. Fig.4, caption: standard deviation of supersaturation fluctuations? 2. L248: When "the" multiplicity. . .

References: Mellado, J.P., Bretherton, C.S., Stevens, B. and Wyant, M.C., 2018. DNS and LES for simulating stratocumulus: better together. Journal of Advances in Modeling Earth Systems, 10(7), pp.1421-1438.
* * *

---

## Referee Comment (RC2) · Anonymous Referee #2 · 14 Apr 2020

This study uses a modified DNS to investigate the effect of turbulence on the diffusional growth of cloud droplets in much larger domains (up to 64*64*64 mˆ3) compared with traditional DNS. The method is the combination of increasing air viscosity to allow DNS in a larger domain and using Lagrangian particle-based microphysics to lower the number of particles needed to be tracked in a larger domain. Results show that the variance of droplet radius increases with the square root of time, which is consistent with previous studies and the result from a stochastic model. Different domain sizes and multiplicities are also tested to check the convergence of the scaled-up DNS. The paper is interesting. However, I have some comments needed to be addressed before this manuscript can be accepted.

Major comments:

1. Running DNS in a larger domain is not new, for example, Rotunno and Bryan (2018) recently applied direct numerical simulation to study lee vortices in a larger domain by setting viscosity as 1 m2/s. Although the combination of this technique and Lagrangian cloud model is interesting, it is not clear to me whether it is suitable to do that, or at least what we can learn from it, for the following reasons:

1.1 When air viscosity is changed, it is unclear to me whether thermal diffusivity is also changed accordingly in this study. If not, the Prandtl number will be different compared with air. It means that this is a fluid that does not behave as air. If yes, this will be consistent with R&B (2018).

1.2 Changing air viscosity and thermal diffusivity will also slow down the condensational growth of cloud droplets following physical rules. But I guess this effect is ignored in this study.

Therefore, the scaled-up DNS in this study simulates a turbulent cloud system that is not similar dimensionless to natural clouds. Therefore, it is unclear to me what we can learn from it. I hope the authors can comment on this.

2. For the same energy dissipation rate, if air viscosity increases, Kolmogorov's length scale increases and therefore a coarser resolution can be used. However, Komogrov's velocity scale also increases at the same time. Larger velocity fluctuation in a larger domain leads to larger supersaturation fluctuation (Figure 7), and thus faster broadening of droplet size distribution (Figure 9). The reason that the variances of droplet size distribution from DNS/scaled-up DNS are consistent with the stochastic model (Figure 10) is that they generate/use the same strength (e.g., PDF) of the velocity field. Therefore, their agreement and the consistency with the $t^{(1/2)}$ scaling law is not a big surprise to me. However, in an idealized homogeneous isotropic turbulent cloud, the velocity (supersaturation) fluctuation should be independent of the volume we choose, meaning that in either a volume of 1 mˆ3 or 10 mˆ3, the energy dissipation rate, Komogrov's length scale and velocity scale should be the same. Therefore, it is unclear to me how the simulated cloud in a domain size associated with the increase of Komogrov's velocity scale, is related to the conceptual cloud with the same large domain with the original (smaller) Komogrov's velocity scale. Please comment on it.

3. Figure 4 shows that results converge for different multiplications. As stated around line 257 "Note that for real DNS (Table 2 and Fig. 4), having a droplet in one of several dozens of grid volumes still results in supersaturation fluctuations in agreement with real droplets." However super droplet even with the multiplicity equals to 1 (real droplets) is one out of eight grid boxes, meaning that the density of droplet is low in the domain. Is it possible the evolution of the supersaturation fluctuation shown in Figure 4 is just the background even without droplet? I think a more careful test is to track more particles in real DNS, at least 10 per grid box, and then change the multiplicity but maintain the same number of particles in the domain.

Minor comments:

1. Line 17: "mean droplet radius variance", should it be "droplet radius variance"?

2. Line 50: "however, see the comment on that paper by. . ." This sentence is not clear to me.

3. Line 101: "K is the molecular diffusion coefficient" should be "thermal diffusion coefficient"

4. Equation 2: I think \delta t should be removed to make sure the unit is correct. Please check.

Reference:

Rotunno, Richard, and George H. Bryan. "Numerical simulations of two-layer flow past topography. Part I: The leeside hydraulic jump." Journal of the Atmospheric Sciences 75, no. 4 (2018): 1231-1241.

---

## Referee Comment (RC3) · Anonymous Referee #2 · 17 Apr 2020

I'd like to thank the authors for their detailed response. I agree with most of their comments except the one below:

The author said that "The molecular coefficients applied in the droplet growth equation are not scaled up and the typical droplet growth equation is used, see below Eq. (2) in the manuscript." However, the growth equation does depend on the coefficients of diffusion and thermal conductivity, which is embedded in the condensational growth equation. For example, check eq. 7.17 in the textbook of Rogers and Yau. In this study, it is assumed to be constant. I am not to say it is forbidden to use a constant. However, justification, clarification, or comments should be added.

[Figure]

In addition, I might not clearly represent my second comment. What I am considering is the following three cases:

A. a small domain (e.g., 1 mˆ3), energy dissipation rate, air viscosity

B. a large domain (e.g., 10 mˆ3), same energy dissipation rate (not TKE), same air viscosity

C. a large domain (e.g., 10 mˆ3), same energy dissipation rate (not TKE), larger air viscosity

In this study, A and C are compared. But I think A and B are more realistic. I know it is impractical to simulate a large domain using DNS without changing the air viscosity. I just curious what if DNS can simulate a large domain (without changing the air viscosity) in the future, is there any difference between the results from B and C the authors expect to see.

In the end, as I said in my comments, this study is interesting, and this technique (DNS in large domain + Lagrangian super droplet) can be applied for other research topics, e.g., as the author mentioned, a rising parcel with CCN activation. But some clarifications related to my comments can help the reader, at least me, to better understand the merits and limitations of this study. I am appreciated for this.

---

## Author Comment (AC1) · 17 Apr 2020

**Responses to comments of the Referee #1.**

Below we respond to the reviewer comments. The original comments are in the black color, our responses blue.

My understanding of the "scaled-DNS" presented in this manuscript is that one can simulate a large domain size by artificially increasing the kinetic viscosity of the airflow. The Reynolds number is kept unchanged in such simulations so that the computational cost is still feasible. Therefore, one can study how the supersaturation fluctuations can be affected by the large eddies. This is plausible as the small scales do not matter for the supersaturation fluctuations. The authors further tested the application of the superdroplet approach in such a setup to tackle the condensation process. I would recommend the publication of this manuscript after the authors carefully discuss the following comments.

The reviewer's summary of our methodology is correct.

The author addressed both in the abstract and in the conclusion that this is a "novel methodology". This statement should be treated carefully for the following reasons: 1. The method presented in this manuscript is DNS with large artificial kinetic viscosity, which is not a new method. 2. Mellado et al also used the same treatment (section 3 of Mellado et al).

We do not agree with this comment. Yes, Mellado et al. applied large domain in their simulations of the stratocumulus top. However, nowhere in Mellado et al. we find the formula showing how the viscosity needs to be increased to allow appropriate dissipation when the model grid length is far away from the dissipation length. This is not that important for the finite-difference model as long as proper dissipation is accomplished by the advection scheme as in the so-called implicit large eddy simulation. But the rescaling is critical for the spectral model that by design has no numerical dissipation. Our methodology targets traditional homogeneous isotropic turbulence simulation applying DNS that in its original form cannot consider droplet growth in large domains. We believe the methodology we propose and use is novel. That said, we added a reference to Mellado et al. and a few other papers (per comments by Rev.1 and 2) in the introduction.

Specific comments:

1. The paper by Mellado et al applied the same idea. Can the authors compare their work with the one of Mellado et al?

Mellado et al. applied similar technique to the cloud top simulation applying bulk condensation scheme. We are not sure what is there to compare.

2. Can the authors check Eq.3 again? If dR/dt=KrS/R, then R^2/2=KrS, together with Eq.2, you will get a pre-factor of 2 instead of 4/3 in Eq.3, right?

No. Eq. 3 does not come from Eq. 2. Eq. 2 is the condensation rate applying the analytic droplet growth rate equation that is shown in the text below Eq. 2. (We corrected an error in Eq. 2 pointed out by Rev. 2). Eq. 3 is the condensation rate derived from the change of the droplet volume as predicted applying a finite time step. Eq. 3 ensures that the mass of water (water vapor plus droplets) is conserved because condensation rate as given by Eq. (3) is applied in the water vapor equation.

3. L.125: Should the intensity of turbulence determined by the single parameter, Reynolds number? The energy dissipate rate is a small-scale quantity which describes how fast energy dissipates in the dissipation range of turbulence. In other words, it characterizes how vigorous the small eddies of turbulence are. It is calculated from the trace of the strain tensor. This aspect is also discussed by the authors in the paragraph just below Eq.10. Did you mean the energy transfer rate here, which is the rate energy transfers from large to small eddies in 3-D turbulence?

Reynolds number describes the range of scales between the scale of energy containing eddies (about the size of the domain in DNS) and the dissipation scale, see Eq. 5 in the paper. The intensity of turbulence is determined by the eddy dissipation rate. Yes, the eddy dissipation rate is a small-scale quantity, but this is how intensity of turbulence is expressed in models and in observations. TKE depends on the eddy dissipation rate and on the characteristic eddy scale, Eq. 4 in the paper. This is why TKE increases with the domain size as shown in Fig. 2 (upper row), and the TKE dissipation rate is the same (lower row).

Perhaps a discussion in Grabowski and Abade (JAS 2017, p. 1485) can also help.

4. Can the authors normalize the energy spectrum in the same way as the one of Fig.1 of Li et al. (2019)? If the Reynolds number is the same, the normalized spectrum should collapse on top of each other.

This can be done. However, we feel the figure in its current form is more informative for a cloud physics reader that we target in our manuscript. The figure clearly shows the shift of spectrum with the increase of L. Also, the figure in the current form illustrates the TKE increases with the domain size for the same TKE dissipation rate. We prefer to leave the figure in its current format.

5. Why is there an initial spike in Fig.6? I don't understand why it is different for different domain sizes. The water mass loading is the same for all the simulations, right?

First, please note that the spike amplitude is very small compared to the standard deviation shown in Fig. 7. We mention that in the revised manuscript. Second, initial conditions are not in the equilibrium with droplets that are present locally, and establishing quasi-equilibrium takes some time. However, one may argue that this should not take as long as the figure shows. Another aspect is that "molecular diffusion" (in quotes as this is the scaled-up molecular diffusion), in addition to the local vertical velocity, impacts droplet growth. In fact, the duration of the spike (longer for larger domains) suggests that the diffusion is likely the key factor. This is where the absence of unresolved scales in the scaled-up DNS may play some role. Because those

transient conditions are just an artifact of the modeling setup, we do not think exploring this any further is warranted.

6. As the integral time scale is different for simulations with different domain size, could it be an idea that the authors normalize the time axis by the integral time scale?

We included two horizontal axes on bottom panels of figure 2 per reviewer request.

Technical corrections:

1. Fig.4, caption: standard deviation of supersaturation fluctuations?

2. L248: When "the" multiplicity. . .

Corrected.

---

## Author Comment (AC2) · 17 Apr 2020

**Responses to comments of the Referee #2.**

Below we respond to the reviewer comments. The original comments are in the black color and our responses are in blue color.

Overall, we feel the reviewer missed the key aspect of the paper: the largest turbulent eddies have the most significant impact on the diffusional droplet growth. Our desire to simulate the large-eddy end of the spectrum, and not scales close to the molecular dissipation, is the main motivation for the scaled-up DNS approach. The motivation has been discussed in the two paragraphs starting at line 64 of the introduction that sets the stage for the entire manuscript.

This study uses a modified DNS to investigate the effect of turbulence on the diffusional growth of cloud droplets in much larger domains (up to 64*64*64 m^3) compared with traditional DNS. The method is the combination of increasing air viscosity to allow DNS in a larger domain and using Lagrangian particle-based microphysics to lower the number of particles needed to be tracked in a larger domain. Results show that the variance of droplet radius increases with the square root of time, which is consistent with previous studies and the result from a stochastic model. Different domain sizes and multiplicities are also tested to check the convergence of the scaled-up DNS. The paper is interesting. However, I have some comments needed to be addressed before this manuscript can be accepted.

This is approximately correct summary of the paper. A small clarification: results from the stochastic model show that the supersaturation fluctuations in the domain are consistent with scaled-up DNS. The figure does not show the droplet radius variance.

Major comments:

1. Running DNS in a larger domain is not new, for example, Rotunno and Bryan (2018) recently applied direct numerical simulation to study lee vortices in a larger domain by setting viscosity as 1 m2/s. Although the combination of this technique and Lagrangian cloud model is interesting, it is not clear to me whether it is suitable to do that, or at least what we can learn from it, for the following reasons:

The lack of novelty comment echoes comments from the Reviewer 1. In general, running a fluid flow model with constant mixing coefficients, such as in Rotunno and Bryan (2018), goes long way back. See, for instance, Grabowski and Clark (JAS 1993, p. 555) for an earlier example. Another example is the so-called implicit large eddy simulation (ILES) approach where there is no explicit mixing and the computational stability is ensured by the numerical algorithm (i.e., the monotone finite-difference advection scheme). Such an approach is being used by many (see, for instance references at the beginning of section 3 in Grabowski JAS 2014, p. 4493). However, ILES would not work for a spectral model because proper dissipation at small scales is missing. In our study, we utilize a spectral DNS code with scaled-up viscosity and diffusivity, and use the code in simulations for domains that are impossible to consider with the traditional DNS. The basis of such an increased dissipation and its application for large domains is the key novelty of our study.

Because of this comment and a comment from the Reviewer 1, we added a couple sentences in the introduction of the revised manuscript.

1.1 When air viscosity is changed, it is unclear to me whether thermal diffusivity is also changed accordingly in this study. If not, the Prandtl number will be different compared with air. It means that this is a fluid that does not behave as air. If yes, this will be consistent with R&B (2018).

Thermal diffusivity is also changed in the same way as the viscosity. This is clearly stated in the abstract and we added a comment on that in section 3 below Eq. (8). The R&B study is irrelevant in this context, see above.

1.2 Changing air viscosity and thermal diffusivity will also slow down the condensational growth of cloud droplets following physical rules. But I guess this effect is ignored in this study. Therefore, the scaled-up DNS in this study simulates a turbulent cloud system that is not similar dimensionless to natural clouds. Therefore, it is unclear to me what we can learn from it. I hope the authors can comment on this.

This comment is incorrect. The molecular coefficients applied in the droplet growth equation are not scaled up and the typical droplet growth equation is used, see below Eq. (2) in the manuscript. The key for the condensational growth are the supersaturation fluctuations. Their magnitude increases with the turbulent eddy size and we aim at producing fluctuations typical for large eddies. The supersaturation fluctuations obtained from scaled-up DNS are consistent with the stochastic model as shown in figure 10. In short, the comment that we simulate a "cloud system that is not similar dimensionless to natural clouds" is simply incorrect.

2. For the same energy dissipation rate, if air viscosity increases, Kolmogorov's length scale increases and therefore a coarser resolution can be used. However, Komogrov's velocity scale also increases at the same time. Larger velocity fluctuation in a larger domain leads to larger supersaturation fluctuation (Figure 7), and thus faster broadening of droplet size distribution (Figure 9). The reason that the variances of droplet size distribution from DNS/scaled-up DNS are consistent with the stochastic model (Figure 10) is that they generate/use the same strength (e.g., PDF) of the velocity field. Therefore, their agreement and the consistency with the t^(1/2) scaling law is not a big surprise to me.

This comment provides an approximately correct summary of our results. However, the reviewer confuses increase of the Kolmogorov scales with the increase of the supersaturation fluctuations. Please see our response to the next comment.

The fact that the results are as expected supports validity of the proposed approach. Yes, it is expected that larger eddies feature larger and longer-lasting supersaturation fluctuations and thus have more impact on the spectral broadening. However, specific details (such as the spectral width increase with the domain size when following t^1/2 scaling, fig. 9) can only be obtained with the scaled-up DNS.

An important comment is that the approach developed in this paper is being used in ongoing studies with more realistic systems, such as a rising parcel with CCN activation. To be

relevant to a turbulent cloud, domains larger than a mere few meters are needed. This aspect is mentioned in the final paragraph of the paper.

However, in an idealized homogeneous isotropic turbulent cloud, the velocity (supersaturation) fluctuation should be independent of the volume we choose, meaning that in either a volume of 1 m^3 or 10 m^3, the energy dissipation rate, Komogrov's length scale and velocity scale should be the same. Therefore, it is unclear to me how the simulated cloud in a domain size associated with the increase of Komogrov's velocity scale, is related to the conceptual cloud with the same large domain with the original (smaller) Komogrov's velocity scale. Please comment on it.

The initial part of the comment is incorrect. The velocity and supersaturation fluctuations **increase** with the domain size when the eddy dissipation (and thus Kolmogorov scales) remain unchanged. This has been shown in past DNS studies that we refer to and is the key of the argument; see also Grabowski and Abade (2017). The largest supersaturation fluctuations come from the largest eddies, and these are related to the turbulent kinetic energy (TKE). This is because TKE represents energy of the largest eddies of a turbulent flow. TKE increases with the domain size and the correct scaling is given by Eq. 4 in the paper (line 129).

3. Figure 4 shows that results converge for different multiplications. As stated around line 257 "Note that for real DNS (Table 2 and Fig. 4), having a droplet in one of several dozens of grid volumes still results in supersaturation fluctuations in agreement with real droplets." However super droplet even with the multiplicity equals to 1 (real droplets) is one out of eight grid boxes, meaning that the density of droplet is low in the domain. Is it possible the evolution of the supersaturation fluctuation shown in Figure 4 is just the background even without droplet? I think a more careful test is to track more particles in real DNS, at least 10 per grid box, and then change the multiplicity but maintain the same number of particles in the domain.

The initial conditions for the simulations are as in Lanotte et al. DNS study, droplet concentration of 130 per cc and droplet radius of 13 microns (see line 119 of our ACPD paper). Indeed, clouds are fairly diluted systems. However, it is not true that the fluctuations for the case with droplets are the same as without droplets. The supersaturation variance in simulations without droplets for the two domain sizes shown in Fig. 7 are about $8 \times 10^{-4}$ for the left panel and about $1.6 \times 10^{-3}$ for the right panel, that is, larger than the mean of those shown in the figure. We do not think we need to discuss this in the paper.

The reviewer's suggestion at the end of the comment does not make sense. Droplet concentration (130 per cc) implies that it is impossible to have 10 droplets per grid box if the grid length is 1 mm. There are on average 0.13 droplets per grid box, or one droplet in about 8 grid boxes as the reviewer notices.

Minor comments:
1. Line 17: "mean droplet radius variance", should it be "droplet radius variance"?

2. Line 50: "however, see the comment on that paper by. . ." This sentence is not clear to me.
3. Line 101: "K is the molecular diffusion coefficient" should be "thermal diffusion coefficient"
4. Equation 2: I think \delta t should be removed to make sure the unit is correct. Please check.

These comments have been addressed in the revision.

---

## Author Comment (AC3) · 18 Apr 2020

Responses to the 2nd comments f the Referee #2.

The reviewer clarifies his/her previous comments. Here are our responses.

1. The issue of the droplet growth equation. As in many studies, we apply a droplet growth equation as $dr/dt = KS/r$, that is, neglecting molecular, curvature, and solute effects. This is justified by the size of droplets we consider. The constant K does depend on molecular water vapor diffusivity and thermal conductivity, as well as local temperature. Lanotte et al. (2009) include such dependencies, but we do not think it

is necessary, see discussion below eq. 1 in Vaillancourt et al. (2001). This is why K is taken as a constant in our simulations. The point is that droplets grow as they do in natural turbulent clouds in response to vertical velocity fluctuations in large eddies, of the scale meters and tens of meters.

2. This comment introduces cases A, B and C. A and B are real DNS with 1 and 10 m**3 domains, respectively, whereas C is the scaled-up DNS with a 10 m**3 domain. A and C are included in the paper, and the reviewer wonders about B. Unfortunately, 10 m**3 domain corresponds to a roughly 2000**3 DNS calculation and it is simply not possible. However, according to scaling in (9) in the paper, B would have about 4.6 higher TKE than A. C, the scaled-up DNS, would have slightly smaller TKE than B. Similar comparison is done for domains of 0.512**3 m**3 and 1.024**3 m**3, see Fig. 3 and its discussion.

We appreciate positive thoughts in the final paragraph of the second comment.

---

## Referee Comment (RC4) · Anonymous Referee #3 · 23 Apr 2020

**General Comments**

This study applies the scaled-up DNS method to simulate supersaturation fluctuations and spectral broadening in an idealized framework of forced, isotropic turbulence. The supersaturation fluctuations are produced by the turbulent vertical motions. Scaled-up DNS is what I would consider to be the simplest possible form of large-eddy simulation. In scaled-up DNS, the molecular diffusivities are increased to maintain the same kinetic energy dissipation rate as an otherwise identical simulation with a smaller grid size. In the present application, the range of scales between the domain size and grid size was maintained. Droplet condensational growth was represented using the superdroplet method. The impact of the droplet multiplicity of the super-droplet method on the simulations was studied. The results (i.e., supersaturation standard deviations) of the scaled-up DNS were compared with those from a stochastic model, and exhibited very good agreement.

**Specific Comments**

1. lines 36-45: What is described here is not the only mechanism by which supersaturation fluctuations can be produced within a cloud, and probably not the msst important. Entrainment and mixing also produce supersaturation fluctuations and are well-known to be an important source of spectral broadening, and should be mentioned here in order to place the focus of this study in proper perspective. As will be mentioned later, the set-up of the simulations actually implies an external forcing, which could be interpreted as a crude representation of entrainment.

2. lines 46-47: It would be appropriate to refer here to the even earlier study by Su et al. (1998) in which diffusional growth of cloud droplets in a turbulent environment was simulated using the Explicit Mixing Parcel Model, which is essentially a 1D kinematic DNS.

   lines 54-56: Droplet sedimentation was included in the EMPM simulations mentioned.

   Su, C.-W., S. K. Krueger, P. A. McMurtry, and P. H. Austin, 1998: Linear eddy modeling of droplet spectral evolution during entrainment and mixing in cumulus clouds. *Atmos. Res.,* **47–48,** 41–58.

3. line 57: Unclear. What has a "small impact on the droplet spectra"?

4. lines 56–64: State what mean supersaturation was used in these studies.

5. line 65: It is possible to do simulations with larger domains with the EMPM. It would appropriate to mention here that the EMPM simulations reported in Su et al. (1998) used a 20-m domain size, and EMPM domains up to 100-m domains were used in Tölle and Krueger (2014).

   Tölle, M. H., and S. K. Krueger, 2014: Effects of entrainment and mixing on the droplet size distributions in warm cumulus clouds. *J. Adv. Model. Earth Syst.,* **6,** 281–299, doi:10.1002/2012MS000209

6. lines 67–69: Be clear about the source of the supersaturation fluctuations that you are referring to, which vertical gradients of potential temperature, not entrainment and mixing. Noting this also makes the explanation of why large eddies produce larger fluctuations more obvious.

7. lines 69–72: It would also be appropriate to mention the EMPM approach here, because it certainly does perform "DNS-like simulations in computational domains comparable to the size of the LES grid box." Furthermore, the recent development of a linear-eddy based SGS model that is combined with the super droplet method by F. Hoffmann should be mentioned.

   Hoffmann, F. and G. Feingold, 2019: Entrainment and Mixing in Stratocumulus: Effects of a New Explicit Subgrid-Scale Scheme for Large-Eddy Simulations with Particle-Based Microphysics. J. Atmos. Sci., 76, 1955-1973, https://doi.org/10.1175/JAS-D-18-0318.1

   Hoffmann, F., T. Yamaguchi, and G. Feingold, 2019: Inhomogeneous Mixing in Lagrangian Cloud Models: Effects on the Production of Precipitation Embryos. J. Atmos. Sci., 76, 113-133, https://doi.org/10.1175/JAS-D-18-0087.1

8. line 100, Eq. (1): In general, this equation should include a term $w'd\bar{T}/dz$. I suspect that this term is missing because $d\bar{T}/dz = 0$ is enforced due to the cyclic b.c. at the top and bottom boundaries. It this is the case, it should be mentioned.

   It should also be mentioned that forcing $d\bar{T}/dz = 0$ is equivalent to forcing a non-zero gradient of potential temperature, which acts as the source of temperature and supersaturation fluctuations.

9. line 111. Eq. (3): State how changes in $R^3$ are calculated if not by using (2).

10. line 118: Please state the initial conditions, particular the initial temperature profile, as well as the initial supersaturation profile.

11. Figure 2: It would be enlightening to the readers to discuss how the TKE can be made non-dimensional in terms of flow parameters (a velocity scale specifically).

12. Figures 6–8: Please state whether the supersaturation statistics plotted are at the droplets or everywhere in the flow. They are most useful for understanding the DSD properties if they are at the droplets, as those for the stochastic model are.

13. Figure 7; It would be enlightening to the readers to discuss how $\sigma_S$ can be made non-dimensional in terms of flow parameters.

14. Figure 9: It would be enlightening to the readers to discuss how $\sigma_{R^2}$ can be made non-dimensional in terms of flow parameters.

15. lines 272-73: Clarify this sentence: "The fluctuating in space supersaturation in the dynamic simulations (i.e., real DNS or scaled-up DNS) is modelled as independent realizations of the fluctuating in time supersaturation." I am not sure what you mean. Modelled by what? The stochastic model? Also clarify that the stochastic model predicts supersaturation fluctuations at the droplets. (See comment 12.)

16. line 283 and Figure 10: Clarify that the stochastic model predicts supersaturation fluctuations at the droplets.

17. line 317: Please explain why small-scale motions would reduce $\sigma_S$.

18. Section 6: It would probably be helpful to many readers to make clear how scaled-up DNS differs from LES. In my view, it is really LES but with a constant eddy viscosity, which is not a good SGS model. Explain why you chose this approach rather than using a better SGS model? I would also suggest that you discuss the advantages and the disadvantages of the scaled-up DNS approach. For the problem addressed, it seems that the stochastic model captures the important physics, and that the scaled-up DNS does not add any additional insights.

19. lines 332-33: I agree with this statement. You may want to mention again other possible methods that have been developed (such as those mentioned in comments 2, 5, and 7).

20. lines 365-6: This might be too general of a statement. The large eddies dominate for this mode of supersaturation fluctuation because they span a larger potential temperature difference for the same mean vertical gradient. For other modes of supersaturation fluctuation generation such as entrainment, large eddies also dominate, but for a different reason (their greater mixing time scale).

21. lines 381-82: "Finally, one can also consider applying scaled-up DNS in simulations of the turbulent entrainment and mixing similar to those discussed in Kumar et al. (2018)." There is a significant drawback for this application of scaled-up DNS due to the importance of the small-scale supersaturation fluctuations in determining DSDs. Such near-Kolmogorov-scale variability is not present in scaled-up DNS.

22.

**Technical Comments**

1. line 139: It would be helpful to define $L_1$ and $L_2$.

---

## Author Comment (AC4) · 27 Apr 2020

**Responses to comments of the Referee #3.**

Below we respond to the reviewer's comments. The original comments are in black color and our responses are in blue. We do not agree with many suggestions as they seem to put our manuscript in a context that we feel is not appropriate for this work. Please see specific responses below.

This study applies the scaled-up DNS method to simulate supersaturation fluctuations and spectral broadening in an idealized framework of forced, isotropic turbulence. The supersaturation fluctuations are produced by the turbulent vertical motions. Scaled-up DNS is what I would consider to be the simplest possible form of large-eddy simulation. In scaled-up DNS, the molecular diffusivities are increased to maintain the same kinetic energy dissipation rate as an otherwise identical simulation with a smaller grid size. In the present application, the range of scales between the domain size and grid size was maintained. Droplet condensational growth was represented using the superdroplet method. The impact of the droplet multiplicity of the super-droplet method on the simulations was studied. The results (i.e., supersaturation standard deviations) of the scaled-up DNS were compared with those from a stochastic model, and exhibited very good agreement.

**Specific Comments**

1. lines 36-45: What is described here is not the only mechanism by which supersaturation fluctuations can be produced within a cloud, and probably not the most important. Entrainment and mixing also produce supersaturation fluctuations and are well-known to be an important source of spectral broadening, and should be mentioned here in order to place the focus of this study in proper perspective. As will be mentioned later, the set-up of the simulations actually implies an external forcing, which could be interpreted as a crude representation of entrainment.

Although we agree in general with the reviewer's comment, we do not think bringing detailed discussion of entrainment and mixing is needed. The paper discusses a very specific aspect of the homogeneous isotropic turbulence impact that was studied in the past applying DNS. Our approach extends those studies and targets DNS community. That said, we modified the introduction and brought several references to entrainment and mixing. We do not understand the last sentence – the simulations are forced in a way typical to traditional DNS.

2. lines 46-47: It would be appropriate to refer here to the even earlier study by Su et al. (1998) in which diffusional growth of cloud droplets in a turbulent environment was simulated using the Explicit Mixing Parcel Model, which is essentially a 1D kinematic DNS. lines 54-56: Droplet sedimentation was included in the EMPM simulations mentioned.

Su, C.-W., S. K. Krueger, P. A. McMurtry, and P. H. Austin, 1998: Linear eddy modeling of droplet spectral evolution during entrainment and mixing in cumulus clouds. Atmos. Res., 47–48, 41–58.

This part of the text has changed. We added several references to entrainment and mixing, including the one suggested. We do not want to single out EMPM as it is only marginally relevant to our study. Please see our response to 1 above.

3. line 57: Unclear. What has a "small impact on the droplet spectra"?

The supersaturation fluctuations. Text modified.

4. lines 56-64: State what mean supersaturation was used in these studies.

Vaillancourt et al. applied a rising adiabatic parcel setup. The mean supersaturation was never presented. Lanotte et al. and Li et al. applied no mean vertical motion (as in our study) and started with zero mean supersaturation. This is now mentioned in the modified text.

5. line 65: It is possible to do simulations with larger domains with the EMPM. It would appropriate to mention here that the EMPM simulations reported in Su et al. (1998) used a 20-m domain size, and EMPM domains up to 100-m domains were used in Tölle and Krueger (2014). Tölle, M. H., and S. K. Krueger, 2014: Effects of entrainment and mixing on the droplet size distributions in warm cumulus clouds. J. Adv. Model. Earth Syst., 6, 281–299, doi:10.1002/2012MS000209

As stated in our response to 1 and 2, we do not want to bring entrainment/mixing in this manuscript except in a brief comment in the final paragraph in the conclusion section.

6. lines 67–69: Be clear about the source of the supersaturation fluctuations that you are referring to, which vertical gradients of potential temperature, not entrainment and mixing. Noting this also makes the explanation of why large eddies produce larger fluctuations more obvious.

There are no mean temperature gradients in the traditional DNS and scaled-up DNS. In the framework we use, larger supersaturations come only from larger and longer-lasting fluctuations of the vertical velocity. We modified the discussion in this paragraph.

7. lines 69–72: It would also be appropriate to mention the EMPM approach here, because it certainly does perform "DNS-like simulations in computational domains comparable to the size of the LES grid box." Furthermore, the recent development of a linear-eddy based SGS model that is combined with the super droplet method by F. Hoffmann should be mentioned. Hoffmann, F. and G. Feingold, 2019: Entrainment and Mixing in Stratocumulus: Effects of a New Explicit Subgrid-Scale Scheme for Large-Eddy Simulations with Particle-Based Microphysics. J. Atmos. Sci., 76, 1955-1973, https://doi.org/10.1175/JAS-D-18-0318.1 Hoffmann, F., T. Yamaguchi, and G. Feingold, 2019: Inhomogeneous Mixing in Lagrangian Cloud Models: Effects on the Production of Precipitation Embryos. J. Atmos. Sci., 76, 113-133, https://doi.org/10.1175/JAS-D-18-0087.1

We do not agree that this aspect needs to be included in the manuscript. We see similarities between DNS and EMPM, but this is only tangentially related to the main thrust of our manuscript.

8. line 100, Eq. (1): In general, this equation should include a term w dT /dz . I suspect that this term is missing because dT /dz = 0 is enforced due to the cyclic b.c. at the top and bottom boundaries. It this is the case, it should be mentioned. It should also be mentioned that forcing dT

/dz = 0 is equivalent to forcing a nonzero gradient of potential temperature, which acts as the source of temperature and supersaturation fluctuations.

The reviewer is correct. DNS by design cannot feature mean temperature gradients because of the triply-periodic boundary conditions. This is why Eq. (1) does not have the w dT/dz term. Eq. (1) is standard for the DNS of homogeneous isotropic turbulence (e.g., see Eq. 9 in Vaillancourt et al. JAS 2001). We prefer not to bring this aspect in the model description.

9. line 111. Eq. (3): State how changes in R3 are calculated if not by using (2).

We added a brief comment on that. The key is that droplet growth is calculated first, and then condensation rate follows from (3).

10. line 118: Please state the initial conditions, particular the initial temperature profile, as well as the initial supersaturation profile.

There are no profiles in the spectral DNS. The initial conditions are specified in the last paragraph of section 2 of the original submission. We added information about the assumed temperature and supersaturation.

11. Figure 2: It would be enlightening to the readers to discuss how the TKE can be made nondimensional in terms of flow parameters (a velocity scale specifically).

We are not sure what the reviewer has in mind here. The velocity scale comes from TKE, see section 5. Perhaps the discussion in Grabowski and Abade (JAS 2017) can help.

12. Figures 6–8: Please state whether the supersaturation statistics plotted are at the droplets or everywhere in the flow. They are most useful for understanding the DSD properties if they are at the droplets, as those for the stochastic model are.

This is a good point. For DNS, the statistics are for the flow and we mention this in the revision. At some point we compared the statistics for the flow and at the droplet positions. The differences between the two methods are small as shown in the figure below, so we decided to use the flow statistics as a much simpler to calculate. We feel this is because gradients at the grid scale are small due to molecular (or scaled-up molecular) transport coefficients and droplets (or super-droplets) present only in the small fraction of grid volumes.

The upper panel in the figure below shows the statistics derived using the flow data. The figure includes some data shown in the manuscript's Fig. 7. The bottom panel shows statistics calculated by interpolating the supersaturation to the droplet position. There are some small differences, but the mean values are close. We mention that in the footnote of the revised section 4.

13. Figure 7; It would be enlightening to the readers to discuss how  $\sigma S$  can be made nondimensional in terms of flow parameters.

We do not think this is possible. There are several parameters that likely play the role, and only some (like the domain size) vary in our study. Perhaps Fig. 10 provides something along the lines suggested by the reviewer.

14. Figure 9: It would be enlightening to the readers to discuss how  $\sigma R2$  can be made nondimensional in terms of flow parameters.

Please see our reply to 13 above.

A general comment to 13 and 14 above: the problem has several parameters. From the fluid flow, the eddy dissipation rate and the domain size are the key (see Grabowski and Abade JAS 2017). From the cloud droplet perspective, the droplet concentration and initial size are relevant. We doubt our limited set of simulations, used primarily to demonstrate the approach strength, allows

to draw specific conclusions in terms of nondimensional parameters. This has to be left for future follow-up studies.

15. lines 272-73: Clarify this sentence: "The fluctuating in space supersaturation in the dynamic simulations (i.e., real DNS or scaled-up DNS) is modelled as independent realizations of the fluctuating in time supersaturation." I am not sure what you mean. Modelled by what? The stochastic model? Also clarify that the stochastic model predicts supersaturation fluctuations at the droplets. (See comment 12.)

The text has been modified to better explain how this is done in the stochastic model. The fact that the stochastic model predicts supersaturation in the droplet vicinity is mentioned in the footnote in section 4.

16. line 283 and Figure 10: Clarify that the stochastic model predicts supersaturation fluctuations at the droplets.

See response to 15.

17. line 317: Please explain why small-scale motions would reduce  $\sigma S$ .

Just simply because of the heuristic argument that missing scales of motions would provide additional smoothing of supersaturation gradients. See also our response to 21 below.

18. Section 6: It would probably be helpful to many readers to make clear how scaledup DNS differs from LES. In my view, it is really LES but with a constant eddy viscosity, which is not a good SGS model. Explain why you chose this approach rather than using a better SGS model? I would also suggest that you discuss the advantages and the disadvantages of the scaled-up DNS approach. For the problem addressed, it seems that the stochastic model captures the important physics, and that the scaled-up DNS does not add any additional insights.

We do not agree with this comment. The paper shows how to expand DNS to larger domains at the same Reynolds number and thus to include larger eddies featuring larger supersaturation fluctuations. We think a better analogy is the so-called implicit LES, that is, an approach used in finite-difference models where no SGS scheme is used. Unfortunately, such a technique cannot be used with a spectral model and this is where the scaled-up DNS fits. We mention this in the revised introduction.

19. lines 332-33: I agree with this statement. You may want to mention again other possible methods that have been developed (such as those mentioned in comments 2, 5, and 7).

Again, we specifically target homogeneous isotropic turbulence simulation methodology. We do not want to bring other methods as not relevant to what we present.

20. lines 365-6: This might be too general of a statement. The large eddies dominate for this mode of supersaturation fluctuation because they span a larger potential temperature difference for the same mean vertical gradient. For other modes of supersaturation fluctuation generation

such as entrainment, large eddies also dominate, but for a different reason (their greater mixing time scale).

This comment is incorrect. Larger eddies feature larger and longer-lasting vertical velocity fluctuations because of the way TKE scales with L for the same eddy dissipation rate. As explained above, spectral DNS has no mean vertical gradients.

21. lines 381-82: "Finally, one can also consider applying scaled-up DNS in simulations of the turbulent entrainment and mixing similar to those discussed in Kumar et al. (2018)." There is a significant drawback for this application of scaledup DNS due to the importance of the small-scale supersaturation fluctuations in determining DSDs. Such near-Kolmogorov-scale variability is not present in scaled-up DNS.

It remains to be seen if the small-scale supersaturation fluctuations are important compared to large-scale fluctuations. The reviewer seems to believe so. We are not sure. Grabowski (JAS 2020) makes that point while discussing ILES simulations of the Pi chamber. He states that only true DNS can answer this question through the comparison with LES or ILES. We added a reference to Paoli and Shariff (2009) that is yet another method to include the impact of entrainment/mixing into DNS and scaled-up DNS in addition to Kumar et al. (2018).

**Technical Comments**

1. line 139: It would be helpful to define L1 and L2.

Text modified.

---

## Referee Report (RR1)

Comments to the revision of manuscript with ID number "acp-2020-159"

I would recommend the publication of this manuscript after the authors carefully discuss the following comments again.

1. I disagree with the response to my comment about the "novel methodology". Again, the method presented in this manuscript is DNS with large artificial kinetic viscosity, which is not a new method.

   Since this is a DNS study, let us put the LES aside. Eq.4-9 in the manuscript show the standard scaling argument of the scale separation of turbulence. DNS (finite difference numerical method or spectral method) of turbulence is limited by Re. To study how large scales of turbulence affect supersaturation fluctuations by means of DNS, the authors increased both the integral length scale and Kolmogorov length scale of turbulence simultaneously so that Re can be kept unchanged. This is very well described by Eq.4-9. What is the novelty regarding numerical methodology in DNS simulations with this configuration? In other words, how can I see the novelty from Eq.4-9?

   The Reynolds number is defined as Re=u_rmsL/\nu. One can simply increase L and \nu at the same time in a DNS simulation to check how large eddies affect supersaturation fluctuations.

   I don't understand the comment about the numerical dissipation for finite difference method and the spectral method. Both methods deal with the physical dissipation, i.e., energy dissipation in the dissipation range of turbulence. How exactly increasing L and \nu simultaneously in a spectral DNS code makes the simulation novel?

   To my knowledge, statistical convergence tests of the superdroplet method in such large simulation domain of DNS (large air viscosity though) has never been done. This is new and useful for the study of diffusional growth of cloud droplets.

2. The authors responded that "Yes, the eddy dissipation rate is a small-scale quantity, but this is how intensity of turbulence is expressed in models and in observations".

   Indeed, in the models and observations, \epsilon has been used to characterize the intensity of turbulence. However, I disagree that turbulence intensity is determined by \epsilon.

   My understanding of the present study is that DNS combined with superdroplet approach is used to study the diffusional growth of cloud droplets. DNS means one solve the Navier-Stokes equation to the native scale of turbulence. Even though Re is small in DNS studies and scale contamination is inevitable, intermittency still exists, i.e., the inhomogeneous distribution of energy dissipation rate in turbulence. This intermittency is determined by the Reynolds number. That is, the energy dissipation rate is determined by the Reynolds number.
   I encourage the authors to have a look at this seminal paper
   https://doi.org/10.1103/PhysRevLett.72.336 and numerous laboratory experimental evidence and observational evidence on this topic.

---

## Referee Report (RR2)

**acp-2020-159 (revised)**

**Title:** Diffusional growth of cloud droplets in homogeneous isotropic turbulence: DNS, scaled-up DNS, and stochastic model

**Authors:** Lois Thomas, Wojciech W. Grabowski, and Bipin Kumar

**General Comments**

I accept most of the authors' responses. However, I do not agree with their responses to the following three comments. I also included two additional comments which are motivated by the authors' responses to some of my other comments.

The original comments are in black, the authors' comments are in blue, and my responses to their responses are in red.

**Specific Comments**

1. line 65: It is possible to do simulations with larger domains with the EMPM. It would appropriate to mention here that the EMPM simulations reported in Su et al. (1998) used a 20-m domain size, and EMPM domains up to 100-m domains were used in Tölle and Krueger (2014).

   Tölle, M. H., and S. K. Krueger, 2014: Effects of entrainment and mixing on the droplet size distributions in warm cumulus clouds. *J. Adv. Model. Earth Syst.,* **6,** 281–299, `doi:10.1002/2012MS000209`

   As stated in our response to 1 and 2, we do not want to bring entrainment/mixing in this manuscript except in a brief comment in the final paragraph in the conclusion section. No changes to the text.

   lines 62-65 (revised): The authors write "From the point of view of realistic cloud modelling, developing and validating robust subgrid-scale schemes for contemporary large eddy simulation (LES) models (i.e., featuring grid lengths of a few tens of meters) requires performing DNS-like simulations in computational domains comparable to the size of the LES grid box." The EMPM does exactly this, as noted in my original comment. This capability is not limited

to entraining parcels. It seems that it would be appropriate to mention the EMPM approach as well. It is clearly relevant to the authors' text.

2. line 100, Eq. (1): In general, this equation should include a term $w'd\bar{T}/dz$. I suspect that this term is missing because $\bar{T}/dz = 0$ is enforced due to the cyclic b.c. at the top and bottom boundaries. It this is the case, it should be mentioned. It should also be mentioned that forcing $\bar{T}/dz = 0$ is equivalent to forcing a non-zero gradient of potential temperature, which acts as the source of temperature and supersaturation fluctuations.

   The reviewer is correct. DNS by design cannot feature mean temperature gradients because of the triply-periodic boundary conditions. This is why Eq. (1) does not have the w dT/dz term. Eq. (1) is standard for the DNS of homogeneous isotropic turbulence (e.g., see Eq. 9 in Vaillancourt et al. JAS 2001). We prefer not to bring this aspect in the model description. No changes to the text.

   The source of the supersaturation fluctuations is vertical velocity fluctuations and condensation. Air parcels ascend or descend along saturated adiabats to a good approximation, so that $dT/dz = \Gamma_s$, which produces temperature fluctuations $\Delta T \approx -\Gamma_s \Delta z$ when $d\bar{T}/dz = 0$. Therefore, the specification of $d\bar{T}/dz = 0$ is important and should be mentioned.

   It also is not true that "DNS by design cannot feature mean temperature gradients". If the thermodynamic variable used in the DNS is potential temperature, for example, then $d\bar{\theta}/dz = 0$ would be required but $d\bar{T}/dz = g/c_p$.

3. lines 365-6: This might be too general of a statement. The large eddies dominate for this mode of supersaturation fluctuation because they span a larger potential temperature difference for the same mean vertical gradient. For other modes of supersaturation fluctuation generation such as entrainment, large eddies also dominate, but for a different reason (their greater mixing time scale).

   This comment is incorrect. Larger eddies feature larger and longer-lasting vertical velocity fluctuations because of the way TKE scales with L for the same eddy dissipation rate. As explained above, spectral DNS has no mean vertical gradients. No changes to the text

   As noted in the previous comment, it is incorrect to state the "spectral DNS has no mean vertical gradients". When $d\bar{T}/dz = 0$, $d\bar{\theta}/dz = \Gamma_d$, for example.

**Addtional Specific Comments**

4. lines 376-383 (revised version): The authors "consider supersaturation fluctuations in a simple stochastic model of a droplet ensemble" and note that "The key advantage of the stochastic model is that its computational cost is just a tiny fraction of a DNS simulation." Furthermore, they write that "the stochastic model provides a simple and physically appealing approach to multiscale large-eddy simulation of a cloud applying Lagrangian particle-based microphysics."

   The same could be said about the EMPM (Su et al. 1998; Tölle, M. H., and S. K. Krueger, 2014) and the L3 model (Hoffmann and Feingold, 2019; Hoffmann et al., 2019). It may benefit the readers to mention these relevant studies.

   > Hoffmann, F. and G. Feingold, 2019: Entrainment and Mixing in Stratocumulus: Effects of a New Explicit Subgrid-Scale Scheme for Large-Eddy Simulations with Particle-Based Microphysics. J. Atmos. Sci., 76, 1955-1973, https://doi.org/10.1175/JAS-D-18-0318.1

   > Hoffmann, F., T. Yamaguchi, and G. Feingold, 2019: Inhomogeneous Mixing in Lagrangian Cloud Models: Effects on the Production of Precipitation Embryos. J. Atmos. Sci., 76, 113-133, https://doi.org/10.1175/JAS-D-18-0087.1

5. lines 384-6 (revised version): The authors write that "The next step can be to apply this approach in a rising parcel simulations..." This statement should be qualified because in the approach described, the supersaturation fluctuations are generated by turbulent vertical motions acting on a specified and unrealistic mean gradient of temperature (isothermal rather than saturated adiabatic). See comment 2.

---

## Author Response (AR2)

**Responses to comments of the Referee #1.**

Below we respond to the reviewer comments. The original comments are in the black color, our responses blue, and additions to the text in red.

My understanding of the "scaled-DNS" presented in this manuscript is that one can simulate a large domain size by artificially increasing the kinetic viscosity of the airflow. The Reynolds number is kept unchanged in such simulations so that the computational cost is still feasible. Therefore, one can study how the supersaturation fluctuations can be affected by the large eddies. This is plausible as the small scales do not matter for the supersaturation fluctuations. The authors further tested the application of the superdroplet approach in such a setup to tackle the condensation process. I would recommend the publication of this manuscript after the authors carefully discuss the following comments.

The reviewer's summary of our methodology is correct.

The author addressed both in the abstract and in the conclusion that this is a "novel methodology". This statement should be treated carefully for the following reasons: 1. The method presented in this manuscript is DNS with large artificial kinetic viscosity, which is not a new method. 2. Mellado et al also used the same treatment (section 3 of Mellado et al).

We do not agree with this comment. Yes, Mellado et al. applied large domain in their simulations of the stratocumulus top. However, nowhere in Mellado et al. we find the formula showing how the viscosity needs to be increased to allow appropriate dissipation when the model grid length is far away from the dissipation length. This is not that important for the finite-difference model as long as proper dissipation is accomplished by the advection scheme as in the so-called implicit large eddy simulation. But the rescaling is critical for the spectral model that by design has no numerical dissipation. Our methodology targets traditional homogeneous isotropic turbulence simulation applying DNS that in its original form cannot consider droplet growth in large domains. We believe the methodology we propose and use is novel. That said, we added a reference to Mellado et al. and a few other papers (per comments by Rev.1 and 2) in the introduction.

The revised paragraph in the introduction now reads (new text in red, but there also some revisions of the wording throughout):

"To this end, we propose to use what we refer to as the "scaled-up DNS" approach. Since the largest eddies are the key for the condensational growth, one would like to apply the DNS technique in simulations with domains much larger than currently possible. For instance, taking a $128^3$ DNS simulation with 10 cm grid length gives computational domain of $12.8^3$ cubic meters, that is, comparable to the grid volume of an LES cloud simulation. To ensure a proper dissipation of the turbulent kinetic energy (TKE) at the smallest scales, one needs to scale up the molecular viscosity with the increase of the model grid length. The increase of the small-scale dissipation is critical for traditional DNS models applying spectral techniques to simulate homogeneous isotropic turbulence as applied in this study. This is different from past turbulence-related studies applying finite-difference models with large domains and spatially-uniform

diffusion coefficients (e.g., Grabowski and Clark, 1993; Mellado et al., 2018; Rotunno and Bryan, 2018) or no explicit diffusion at all as in the so-called implicit large eddy simulation (ILES; e.g., Margolin et al., 2006; Grinstein et al., 2007). The increased number of droplets in the large domain can be accounted for by the so-called super-droplet technique in which each super-droplet represents an appropriately scaled-up number of natural droplets (referred to as the multiplicity factor, Shima et al., 2009) as already applied in the appendix of Lanotte et al. (2009) and in Li et al. (2019).”

Included (in red) at the beginning of section 3: “With the increased grid length, one needs to increase molecular transport coefficients to maintain proper TKE dissipation as well as the removal of scalar fluctuations. Note that this is different from LES and ILES methodologies mentioned in the introduction and provides the key novelty for the spectral homogeneous isotropic turbulence simulation. Assuming that the domain size…”.

Specific comments:

1. The paper by Mellado et al applied the same idea. Can the authors compare their work with the one of Mellado et al?

 Mellado et al. applied similar technique to the cloud top simulation applying bulk condensation scheme. We are not sure what is there to compare.

No changes to the text.

2. Can the authors check Eq.3 again? If dR/dt=KrS/R, then R^2/2=KrS, together with Eq.2, you will get a pre-factor of 2 instead of 4/3 in Eq.3, right?

No. Eq. 3 does not come from Eq. 2. Eq. 2 is the condensation rate applying the analytic droplet growth rate equation that is shown in the text below Eq. 2. (We corrected an error in Eq. 2 pointed out by Rev. 2). Eq. 3 is the condensation rate derived from the change of the droplet volume as predicted applying a finite time step. Eq. 3 ensures that the mass of water (water vapor plus droplets) is conserved because condensation rate as given by Eq. (3) is applied in the water vapor equation.

Added below eq. (3): “This means that droplet growth is calculated first, and then (3) is used to derive the condensation rate.”

3. L.125: Should the intensity of turbulence determined by the single parameter, Reynolds number? The energy dissipate rate is a small-scale quantity which describes how fast energy dissipates in the dissipation range of turbulence. In other words, it characterizes how vigorous the small eddies of turbulence are. It is calculated from the trace of the strain tensor. This aspect is also discussed by the authors in the paragraph just below Eq.10. Did you mean the energy transfer rate here, which is the rate energy transfers from large to small eddies in 3-D turbulence?

Reynolds number describes the range of scales between the scale of energy containing eddies (about the size of the domain in DNS) and the dissipation scale, see Eq. 5 in the paper. The intensity of turbulence is determined by the eddy dissipation rate. Yes, the eddy dissipation rate is a small-scale quantity, but this is how intensity of turbulence is expressed in models and in observations. TKE depends on the eddy dissipation rate and on the characteristic eddy scale, Eq. 4 in the paper. This is why TKE increases with the domain size as shown in Fig. 2 (upper row), and the TKE dissipation rate is the same (lower row). Perhaps the discussion in Grabowski and Abade (JAS 2017, p. 1485) can also help.

No change to the text.

4. Can the authors normalize the energy spectrum in the same way as the one of Fig.1 of Li et al. (2019)? If the Reynolds number is the same, the normalized spectrum should collapse on top of each other.

This can be done. However, we feel the figure in its current form is more informative for a cloud physics reader that we target in our manuscript. The figure clearly shows the shift of spectrum with the increase of L. Also, the figure in the current form illustrates the TKE increases with the domain size for the same TKE dissipation rate. We prefer to leave the figure in its current format.

No change to the text and figure.

5. Why is there an initial spike in Fig.6? I don't understand why it is different for different domain sizes. The water mass loading is the same for all the simulations, right?

First, please note that the spike amplitude is very small compared to the standard deviation shown in Fig. 7. We mention that in the revised manuscript. Second, initial conditions are not in the equilibrium with droplets that are present locally, and establishing quasi-equilibrium takes some time. However, one may argue that this should not take as long as the figure shows. Another aspect is that "molecular diffusion" (in quotes as this is the scaled-up molecular diffusion), in addition to the local vertical velocity, impacts droplet growth. In fact, the duration of the spike (longer for larger domains) suggests that the diffusion is likely the key factor. This is where the absence of unresolved scales in the scaled-up DNS may play some role. Because those transient conditions are just an artifact of the modeling setup, we do not think exploring this any further is warranted.

Added to the text in red: "The spike magnitude, about hundred times smaller than the standard deviations shown in Fig. 7, increases as the domain size increases…".

6. As the integral time scale is different for simulations with different domain size, could it be an idea that the authors normalize the time axis by the integral time scale?

We included two horizontal axes on bottom panels of figure 2 per reviewer request.

Technical corrections:

1. Fig.4, caption: standard deviation of supersaturation fluctuations?

2. L248: When "the" multiplicity. . .

Corrected as suggested.

**Responses to comments of the Referee #2.**

Below we respond to the reviewer comments. The original comments are in the black color and our responses are in blue color. New text is in red.

Overall, we feel the reviewer missed the key aspect of the paper: the largest turbulent eddies have the most significant impact on the diffusional droplet growth. Our desire to simulate the large-eddy end of the spectrum, and not scales close to the molecular dissipation, is the main motivation for the scaled-up DNS approach. The motivation has been discussed in the two paragraphs starting at line 64 of the introduction that sets the stage for the entire manuscript.

This study uses a modified DNS to investigate the effect of turbulence on the diffusional growth of cloud droplets in much larger domains (up to 64*64*64 mˆ3) compared with traditional DNS. The method is the combination of increasing air viscosity to allow DNS in a larger domain and using Lagrangian particle-based microphysics to lower the number of particles needed to be tracked in a larger domain. Results show that the variance of droplet radius increases with the square root of time, which is consistent with previous studies and the result from a stochastic model. Different domain sizes and multiplicities are also tested to check the convergence of the scaled-up DNS. The paper is interesting. However, I have some comments needed to be addressed before this manuscript can be accepted.

This is approximately correct summary of the paper. A small clarification: results from the stochastic model show that the supersaturation fluctuations in the domain are consistent with scaled-up DNS. The figure does not show the droplet radius variance.

Major comments:

1. Running DNS in a larger domain is not new, for example, Rotunno and Bryan (2018) recently applied direct numerical simulation to study lee vortices in a larger domain by setting viscosity as 1 m2/s. Although the combination of this technique and Lagrangian cloud model is interesting, it is not clear to me whether it is suitable to do that, or at least what we can learn from it, for the following reasons:

The lack of novelty comment echoes comments from the Reviewer 1. In general, running a fluid flow model with constant mixing coefficients, such as in Rotunno and Bryan (2018), goes long way back. See, for instance, Grabowski and Clark (JAS 1993, p. 555) for an earlier example. Another example is the so-called implicit large eddy simulation (ILES) approach where there is no explicit mixing and the computational stability is ensured by the numerical algorithm (i.e., the monotone finite-difference advection scheme). Such an approach is being used by many (see, for instance references at the beginning of section 3 in Grabowski JAS 2014, p. 4493). However, ILES would not work for a spectral model because proper dissipation at small scales is missing. In our study, we utilize a spectral DNS code with scaled-up viscosity and diffusivity, and use the code in simulations for domains that are impossible to consider with the traditional DNS. The basis of such an increased dissipation and its application for large domains is the key novelty of our study.

Because of this comment and a comment from the Reviewer 1, we added a sentence in the introduction of the revised manuscript. The revised paragraph in the introduction now reads (new text in red, but there also small revisions of the wording throughout):

"To this end, we propose to use what we refer to as the "scaled-up DNS" approach. Since the largest eddies are the key for the condensational growth, one would like to apply the DNS technique in simulations with domains much larger than currently possible. For instance, taking a $128^3$ DNS simulation with 10 cm grid length gives computational domain of $12.8^3$ cubic meters, that is, comparable to the grid volume of an LES cloud simulation. To ensure a proper dissipation of the turbulent kinetic energy (TKE) at the smallest scales, one needs to scale up the molecular viscosity with the increase of the model grid length. The increase of the small-scale dissipation is critical for traditional DNS models applying spectral techniques to simulate homogeneous isotropic turbulence as applied in this study. This is different from past turbulence-related studies applying finite-difference models with large domains and spatially-uniform diffusion coefficients (e.g., Grabowski and Clark, 1993; Mellado et al., 2018; Rotunno and Bryan, 2018) or no explicit diffusion at all as in the so-called implicit large eddy simulation (ILES; e.g., Margolin et al., 2006; Grinstein et al., 2007). The increased number of droplets in the large domain can be accounted for by the so-called super-droplet technique in which each super-droplet represents an appropriately scaled-up number of natural droplets (referred to as the multiplicity factor, Shima et al., 2009) as already applied in the appendix of Lanotte et al. (2009) and in Li et al. (2019)."

Included (in red) at the beginning of section 3: "With the increased grid length, one needs to increase molecular transport coefficients to maintain proper TKE dissipation as well as the removal of scalar fluctuations. Note that this is different from LES and ILES methodologies mentioned in the introduction and provides the key novelty for the spectral homogeneous isotropic turbulence simulation. Assuming that the domain size…".

1.1 When air viscosity is changed, it is unclear to me whether thermal diffusivity is also changed accordingly in this study. If not, the Prandtl number will be different compared with air. It means that this is a fluid that does not behave as air. If yes, this will be consistent with R&B (2018).

Thermal diffusivity is also changed in the same way as the viscosity. This is clearly stated in the abstract and we added a comment on that in section 3 below Eq. (8). The R&B study is irrelevant in this context, see above.

Added below Eq. (8): "Thermal and water vapor diffusivities are also changed in the same way as the viscosity.".

1.2 Changing air viscosity and thermal diffusivity will also slow down the condensational growth of cloud droplets following physical rules. But I guess this effect is ignored in this study. Therefore, the scaled-up DNS in this study simulates a turbulent cloud system that is not similar dimensionless to natural clouds. Therefore, it is unclear to me what we can learn from it. I hope the authors can comment on this.

This comment is incorrect. The molecular coefficients applied in the droplet growth equation are not scaled up and the typical droplet growth equation is used, see below Eq. (2) in the manuscript. The key for the condensational growth are the supersaturation fluctuations. Their magnitude increases with the turbulent eddy size and we aim at producing fluctuations typical for large eddies. The supersaturation fluctuations obtained from scaled-up DNS are consistent with the stochastic model as shown in figure 10. In short, the comment that we simulate a "cloud system that is not similar dimensionless to natural clouds" is simply incorrect.

No change to the text.

2. For the same energy dissipation rate, if air viscosity increases, Kolmogorov's length scale increases and therefore a coarser resolution can be used. However, Komogrov's velocity scale also increases at the same time. Larger velocity fluctuation in a larger domain leads to larger supersaturation fluctuation (Figure 7), and thus faster broadening of droplet size distribution (Figure 9). The reason that the variances of droplet size distribution from DNS/scaled-up DNS are consistent with the stochastic model (Figure 10) is that they generate/use the same strength (e.g., PDF) of the velocity field. Therefore, their agreement and the consistency with the $t^{(1/2)}$ scaling law is not a big surprise to me.

This comment provides an approximately correct summary of our results.

The fact that the results are as expected supports validity of the proposed approach. Yes, it is expected that larger eddies feature larger and longer-lasting supersaturation fluctuations and thus have more impact on the spectral broadening. However, specific details (such as the spectral width increase with the domain size when following $t^{1/2}$ scaling, fig. 9) can only be obtained with the scaled-up DNS.

An important comment is that the approach developed in this paper is being used in ongoing studies with more realistic systems, such as a rising parcel with CCN activation. To be relevant to a turbulent cloud, domains larger than a mere few meters are needed. This aspect is mentioned in the final paragraph of the paper.

No changes to the text.

 However, in an idealized homogeneous isotropic turbulent cloud, the velocity (supersaturation) fluctuation should be independent of the volume we choose, meaning that in either a volume of 1 m^3 or 10 m^3, the energy dissipation rate, Komogrov's length scale and velocity scale should be the same. Therefore, it is unclear to me how the simulated cloud in a domain size associated with the increase of Komogrov's velocity scale, is related to the conceptual cloud with the same large domain with the original (smaller) Komogrov's velocity scale. Please comment on it.

The initial part of the comment is incorrect. The velocity and supersaturation fluctuations **increase** with the domain size when the eddy dissipation (and thus Kolmogorov scales) remain unchanged. This has been shown in past DNS studies that we refer to and is the key

of the argument; see also Grabowski and Abade (2017). The largest supersaturation fluctuations come from the largest eddies, and these are related to the turbulent kinetic energy (TKE). This is because TKE represents energy of the largest eddies of a turbulent flow. TKE increases with the domain size and the correct scaling is given by Eq. 4 in the paper.

No change to the text.

3. Figure 4 shows that results converge for different multiplications. As stated around line 257 "Note that for real DNS (Table 2 and Fig. 4), having a droplet in one of several dozens of grid volumes still results in supersaturation fluctuations in agreement with real droplets." However super droplet even with the multiplicity equals to 1 (real droplets) is one out of eight grid boxes, meaning that the density of droplet is low in the domain. Is it possible the evolution of the supersaturation fluctuation shown in Figure 4 is just the background even without droplet? I think a more careful test is to track more particles in real DNS, at least 10 per grid box, and then change the multiplicity but maintain the same number of particles in the domain.

The initial conditions for the simulations are as in Lanotte et al. DNS study, droplet concentration of 130 per cc and droplet radius of 13 microns (see line 119 of our ACPD paper). Indeed, clouds are fairly diluted systems. However, it is not true that the fluctuations for the case with droplets are the same as without droplets. The supersaturation variance in simulations without droplets for the two domain sizes shown in Fig. 7 are about $8 \times 10^{-4}$ for the left panel and about $1.6 \times 10^{-3}$ for the right panel, that is, larger than the mean of those shown in the figure. We do not think we need to discuss this in the paper. Also, please see revised Fig. 10.

The reviewer's suggestion at the end of the comment does not make sense. Droplet concentration (130 per cc) implies that it is impossible to have 10 droplets per grid box if the grid length is 1 mm. There are on average 0.13 droplets per grid box, or one droplet in about 8 grid boxes as the reviewer notices.

Changes to the paper: Figure 10 has been revised. It now shows that supersaturations without droplets are larger compared to the case with droplets, see the red line.

Minor comments:
1. Line 17: "mean droplet radius variance", should it be "droplet radius variance"?
   "Mean" was removed..
2. Line 50: "however, see the comment on that paper by. . ." This sentence is not clear to me.
   This sentence has been removed.
2. Line 101: "K is the molecular diffusion coefficient" should be "thermal diffusion coefficient"
   Changed as suggested.
3. Equation 2: I think \delta t should be removed to make sure the unit is correct. Please check.

Yes, this was a typo. Corrected.

**Responses to comments of the Referee #3.**

Below we respond to the reviewer's comments. The original comments are in black, our responses are in blue, and additions to the text are in red. We do not agree with many suggestions. We feel those suggestions put our manuscript in the context that we feel is not appropriate for this work. Nether the entrainment/mixing nor the LES aspect is relevant for this work. Please see specific responses below.

This study applies the scaled-up DNS method to simulate supersaturation fluctuations and spectral broadening in an idealized framework of forced, isotropic turbulence. The supersaturation fluctuations are produced by the turbulent vertical motions. Scaled-up DNS is what I would consider to be the simplest possible form of large-eddy simulation. In scaled-up DNS, the molecular diffusivities are increased to maintain the same kinetic energy dissipation rate as an otherwise identical simulation with a smaller grid size. In the present application, the range of scales between the domain size and grid size was maintained. Droplet condensational growth was represented using the superdroplet method. The impact of the droplet multiplicity of the super-droplet method on the simulations was studied. The results (i.e., supersaturation standard deviations) of the scaled-up DNS were compared with those from a stochastic model, and exhibited very good agreement.

Specific Comments

1. lines 36-45: What is described here is not the only mechanism by which supersaturation fluctuations can be produced within a cloud, and probably not the most important. Entrainment and mixing also produce supersaturation fluctuations and are well-known to be an important source of spectral broadening, and should be mentioned here in order to place the focus of this study in proper perspective. As will be mentioned later, the set-up of the simulations actually implies an external forcing, which could be interpreted as a crude representation of entrainment.

Although we agree in general with the reviewer's comment, we do not think bringing detailed discussion of entrainment and mixing is needed. The paper discusses a very specific aspect of the homogeneous isotropic turbulence impact that was studied in the past applying DNS. Our approach extends those studies and targets DNS community. That said, we modified the introduction and brought several references to entrainment and mixing. We do not understand the last sentence – the simulations are forced in a way typical to the traditional DNS model.

In the revised introduction, the paragraph starting with "Srivastava (1989) was first…" was removed. The revised initial paragraph in the introduction reads (added text in red):

"The impact of turbulence on the growth of cloud droplets is an important and still poorly understood aspect of cloud physics. This is because of the wide range of spatial scales that affect droplet growth, from the Kolmogorov microscale (about a millimeter for typical atmospheric turbulence levels) to the scale of the entire cloud or cloud system. Cloud droplets grow by the diffusion of water vapor and by gravitational collision/coalescence, with the former dominating growth until droplets are large enough so the collisional growth can be initiated and eventually led to drizzle and rain formation. For the gravitational collision/coalescence, the frequency of droplet collisions depends on the droplet spectrum width. It follows that understanding processes

leading to the observed droplet spectra is important for the understanding of the rain onset. Observations of natural droplet spectra go back to the early days of aircraft cloud studies (e.g., Warner, 1969) and continue in numerous subsequent investigations (e.g., Jensen et al., 1985; Brenguier and Chaumat, 2001; Pawlowska et al., 2006; Prabha et al., 2012, among many others; see also references in Grabowski and Wang, 2013). Those observations typically show that observed droplet spectra are wider than predicted by simple models of cloud dynamics and microphysics. In many instances, such a discrepancy can be explained by cloud entrainment (e.g., Warner, 1973; Paluch and Knight, 1984; Su et al., 1998; Lasher-Trapp et al., 2005, among many others). However, presence of a significant spectral broadening in undiluted and weakly diluted cloudy volumes is more difficult to explain. One can wonder if the presence of small-scale turbulence can lead to appreciable widening of the droplet spectra during diffusional growth within otherwise uniform cloudy volumes."

There are also additional revisions of the introduction as explained in responses to Rev 1 and 2, and documented in the pdf difference file attached to our responses.

2. lines 46-47: It would be appropriate to refer here to the even earlier study by Su et al. (1998) in which diffusional growth of cloud droplets in a turbulent environment was simulated using the Explicit Mixing Parcel Model, which is essentially a 1D kinematic DNS. lines 54-56: Droplet sedimentation was included in the EMPM simulations mentioned.
Su, C.-W., S. K. Krueger, P. A. McMurtry, and P. H. Austin, 1998: Linear eddy modeling of droplet spectral evolution during entrainment and mixing in cumulus clouds. Atmos. Res., 47–48, 41–58.

This part of the text has changed. We added several references to entrainment and mixing, including the one suggested, see above. We do not want to single out EMPM as it is only marginally relevant to our study. Please see our response to 1 above.

3. line 57: Unclear. What has a "small impact on the droplet spectra"?

The supersaturation fluctuations. Text modified: "Vaillancourt et al. (2002) simulations show a small impact of local supersaturation fluctuations on the droplet spectra…"

4. lines 56–64: State what mean supersaturation was used in these studies.

Vaillancourt et al. applied a rising adiabatic parcel setup. The mean supersaturation was never presented. Lanotte et al. and Li et al. applied no mean vertical motion (as in our study) and started with zero mean supersaturation. This is now mentioned in the modified text:

"Vaillancourt et al. (2001, 2002) were first to apply direct numerical simulation (DNS) approach to study diffusional growth of cloud droplets in homogeneous isotropic turbulence applying a rising adiabatic parcel setup. (…) Similar simulations reported in Lanotte et al. (2009) applying larger domains and no mean ascent clearly show…"

5. line 65: It is possible to do simulations with larger domains with the EMPM. It would appropriate to mention here that the EMPM simulations reported in Su et al. (1998) used a 20-m

domain size, and EMPM domains up to 100-m domains were used in Tölle and Krueger (2014). Tölle, M. H., and S. K. Krueger, 2014: Effects of entrainment and mixing on the droplet size distributions in warm cumulus clouds. J. Adv. Model. Earth Syst., 6, 281–299, doi:10.1002/2012MS000209

As stated in our response to 1 and 2, we do not want to bring entrainment/mixing in this manuscript except in a brief comment in the final paragraph in the conclusion section.

No changes to the text.

6. lines 67–69: Be clear about the source of the supersaturation fluctuations that you are referring to, which vertical gradients of potential temperature, not entrainment and mixing. Noting this also makes the explanation of why large eddies produce larger fluctuations more obvious.

There are no mean temperature gradients in the traditional DNS and scaled-up DNS. In the framework we use, larger supersaturations come only from larger and longer-lasting fluctuations of the vertical velocity. We modified the discussion in this paragraph to read:

"Homogeneous isotropic turbulence simulations of Vaillancourt et al. (2002), Lanotte et al. (2009) and Li et al. (2019) are limited by the computational domain size. As a result, simulations featuring domains larger than a fraction of a cubic meter are simply not yet possible. At the same time, as argued in Grabowski and Wang (2013) and documented in Grabowski and Abade (2017; see Fig. 4 therein) and Li et al. (2019; see Fig. 4 therein), the impact of supersaturation fluctuations in homogeneous isotropic turbulence on the spectral width increases with the domain size. A simple argument is that this is because the largest turbulent eddies feature the largest vertical velocity perturbations that result in the largest and longest-lasting supersaturation fluctuations and thus have the largest impact on the spread of droplet growth histories. From the point of view of realistic cloud modelling, developing and validating robust subgrid-scale schemes for contemporary large eddy simulation (LES) models (i.e., featuring grid lengths of a few tens of meters) requires performing DNS-like simulations in computational domains comparable to the size of the LES grid box."

7. lines 69–72: It would also be appropriate to mention the EMPM approach here, because it certainly does perform "DNS-like simulations in computational domains comparable to the size of the LES grid box." Furthermore, the recent development of a linear-eddy based SGS model that is combined with the super droplet method by F. Hoffmann should be mentioned. Hoffmann, F. and G. Feingold, 2019: Entrainment and Mixing in Stratocumulus: Effects of a New Explicit Subgrid-Scale Scheme for Large-Eddy Simulations with Particle-Based Microphysics. J. Atmos. Sci., 76, 1955-1973, https://doi.org/10.1175/JAS-D-18-0318.1 Hoffmann, F., T. Yamaguchi, and G. Feingold, 2019: Inhomogeneous Mixing in Lagrangian Cloud Models: Effects on the Production of Precipitation Embryos. J. Atmos. Sci., 76, 113-133, https://doi.org/10.1175/JAS-D-18-0087.1

We do not agree that this aspect needs to be included in the manuscript. We see similarities between DNS and EMPM, but this is only tangentially related to the main thrust of our manuscript.

No changes to the text.

8. line 100, Eq. (1): In general, this equation should include a term w dT /dz . I suspect that this term is missing because dT /dz ¯ = 0 is enforced due to the cyclic b.c. at the top and bottom boundaries. It this is the case, it should be mentioned. It should also be mentioned that forcing dT /dz = 0 is equivalent to forcing a nonzero gradient of potential temperature, which acts as the source of temperature and supersaturation fluctuations.

The reviewer is correct. DNS by design cannot feature mean temperature gradients because of the triply-periodic boundary conditions. This is why Eq. (1) does not have the w dT/dz term. Eq. (1) is standard for the DNS of homogeneous isotropic turbulence (e.g., see Eq. 9 in Vaillancourt et al. JAS 2001). We prefer not to bring this aspect in the model description.

No changes to the text.

9. line 111. Eq. (3): State how changes in R3 are calculated if not by using (2).

We added a brief comment on that. The key is that droplet growth is calculated first, and then condensation rate follows from (3).

Added below eq. (3): "This means that droplet growth is calculated first, and then (3) is used to derive the condensation rate."

10. line 118: Please state the initial conditions, particular the initial temperature profile, as well as the initial supersaturation profile.

There are no profiles in the spectral DNS. The initial conditions are specified in the last paragraph of section 2 of the original submission. We added information about the assumed temperature and supersaturation. The revised text reads (new in red):

"The modeling setup follows one of the simulations discussed in Lanotte et al. (2009). We consider an initial mono-disperse droplet distribution of 13 μm radius and the concentration of 130 cm−3. The liquid water content (LWC) is 1.19 gm−3. The initial conditions include uniform temperature of 283 K and zero supersaturation. The later gives the water vapor mixing ratio of 7.65 gkg−1. Since the mean velocity inside the DNS domain is zero, the total cloud water does not change with time, but the initial monodisperse droplet size distribution broadens because the supersaturation fluctuates in time and space affecting the distribution (cf. Li et al., 2019; Saito et al., 2019). The two specific aspects are discussed in the next two section that allow extending the DNS methodology to apply large spatial domains.'

11. Figure 2: It would be enlightening to the readers to discuss how the TKE can be made non-dimensional in terms of flow parameters (a velocity scale specifically).

We are not sure what the reviewer has in mind here. The velocity scale comes from TKE, see section 5. Perhaps the discussion in Grabowski and Abade (JAS 2017) can help.

No change to the text.

12. Figures 6–8: Please state whether the supersaturation statistics plotted are at the droplets or everywhere in the flow. They are most useful for understanding the DSD properties if they are at the droplets, as those for the stochastic model are.

This is a good point. For DNS, the statistics are for the flow and we mention this in the revision. At some point we compared the statistics for the flow and at the droplet positions. The differences between the two methods are small as shown in the figure below, so we decided to use the flow statistics as a much simpler to calculate. We feel this is because gradients at the grid scale are small due to molecular (or scaled-up molecular) transport coefficients and droplets (or super-droplets) present only in the small fraction of grid volumes.

The upper panel in the figure below shows the statistics derived using the flow data. The figure includes some data shown in the manuscript's Fig. 7. The bottom panel shows statistics calculated by interpolating the supersaturation to the droplet position. There are some small differences, but the mean values are close. We mention that in the footnote of the revised section 4.

[Figure]

The added footnote reads: "Supersaturation statistics in DNS and scaled-up DNS are calculated using fluid flow grid data and not the supersaturation interpolated to droplet positions. Limited tests suggest that the differences between the two methods are small (not shown).

Supersaturation statistics for the stochastic model in Section 5 are for the vicinity of a droplet. Discussion in the Appendix A of Vaillancourt et al. (2001) is pertinent to this issue."

13. Figure 7; It would be enlightening to the readers to discuss how σS can be made non-dimensional in terms of flow parameters.

We do not think this is possible. There are several parameters that likely play the role, and only some (like the domain size) vary in our study. Perhaps Fig. 10 provides something along the lines suggested by the reviewer.

No change to the text.

14. Figure 9: It would be enlightening to the readers to discuss how σR2 can be made non-dimensional in terms of flow parameters.

Please see our reply to 13 above. No change to the text.

A general comment to 13 and 14 above: the problem has several parameters. From the fluid flow, the eddy dissipation rate and the domain size are the key (see Grabowski and Abade JAS 2017). From the cloud droplet perspective, the droplet concentration and initial size are relevant. We doubt our limited set of simulations, used primarily to demonstrate the approach, allows to draw specific conclusions in terms of nondimensional parameters. This has to be left for future follow-up studies.

15. lines 272-73: Clarify this sentence: "The fluctuating in space supersaturation in the dynamic simulations (i.e., real DNS or scaled-up DNS) is modelled as independent realizations of the fluctuating in time supersaturation." I am not sure what you mean. Modelled by what? The stochastic model? Also clarify that the stochastic model predicts supersaturation fluctuations at the droplets. (See comment 12.)

The text has been modified to better explain how this is done in the stochastic model. The fact that the stochastic model predicts supersaturation in the droplet vicinity is mentioned in the footnote in section 4. The revised text reads:

"The fluctuating in space supersaturation in the dynamic simulations (i.e., real DNS or scaled-up DNS) is modelled in the stochastic model as independent realizations of the fluctuating in time supersaturation as described below."

16. line 283 and Figure 10: Clarify that the stochastic model predicts supersaturation fluctuations at the droplets.

This is mentioned in the footnote (see 15 above). The figure was modified in the revision.

17. line 317: Please explain why small-scale motions would reduce σS.

Just simply because of the heuristic argument that missing scales of motions would provide additional smoothing of supersaturation gradients. See also our response to 21 below.

18. Section 6: It would probably be helpful to many readers to make clear how scaledup DNS differs from LES. In my view, it is really LES but with a constant eddy viscosity, which is not a good SGS model. Explain why you chose this approach rather than using a better SGS model? I would also suggest that you discuss the advantages and the disadvantages of the scaled-up DNS approach. For the problem addressed, it seems that the stochastic model captures the important physics, and that the scaled-up DNS does not add any additional insights.

We do not agree with this comment. The paper shows how to expand traditional DNS to larger domains at the same Reynolds number and thus to include larger eddies featuring larger supersaturation fluctuations. We think a better analogy is the so-called implicit LES (ILES), that is, an approach used in finite-difference models where no SGS scheme is used. Unfortunately, such a technique cannot be used with a spectral model and this is where the scaled-up DNS fits. We mention this in the revised introduction. For the second half of the comment, we think the last paragraph of the conclusion section provides specific science problems to which scaled-up DNS can be applied.

No changes to the text except of additional reference, Paoli and Shariff (2009).

19. lines 332-33: I agree with this statement. You may want to mention again other possible methods that have been developed (such as those mentioned in comments 2, 5, and 7).

Again, we specifically target homogeneous isotropic turbulence simulation methodology. We do not want to bring other methods as not relevant to what we present.

No changes to the text.

20. lines 365-6: This might be too general of a statement. The large eddies dominate for this mode of supersaturation fluctuation because they span a larger potential temperature difference for the same mean vertical gradient. For other modes of supersaturation fluctuation generation such as entrainment, large eddies also dominate, but for a different reason (their greater mixing time scale).

This comment is incorrect. Larger eddies feature larger and longer-lasting vertical velocity fluctuations because of the way TKE scales with L for the same eddy dissipation rate. As explained above, spectral DNS has no mean vertical gradients.

No changes to the text

21. lines 381-82: "Finally, one can also consider applying scaled-up DNS in simulations of the turbulent entrainment and mixing similar to those discussed in Kumar et al. (2018)." There is a significant drawback for this application of scaledup DNS due to the importance of the small-scale supersaturation fluctuations in determining DSDs. Such near-Kolmogorov-scale variability is not present in scaled-up DNS.

It remains to be seen if the small-scale supersaturation fluctuations are important compared to large-scale fluctuations. The reviewer seems to believe so. We are not sure. Grabowski (JAS 2020) makes that point while discussing ILES simulations of the Pi chamber. He states that only true DNS can answer this question through the comparison with LES or ILES.

We added a reference to Paoli and Shariff (2009) that is yet another method to include the impact of entrainment/mixing into DNS and scaled-up DNS in addition to Kumar et al. (2018).

Technical Comments

1. line 139: It would be helpful to define L1 and L2.

[revised manuscript text omitted]

---

## Author Response (AR3)

**Responses to comments of the Referee #2 and #3.**

Below we respond to the reviewers' comments. For #2, the comments are in the black, our responses in green. For #2, we retain colors used by the Reviewer and use green in our responses.

**Referee #2**

I would recommend the publication of this manuscript after the authors carefully discuss the following comments again.

1. I disagree with the response to my comment about the "novel methodology". Again, the method presented in this manuscript is DNS with large artificial kinetic viscosity, which is not a new method.

The word "novel" is used 4 times in the text: twice in the abstract, then in line 133, and at the beginning of the summary section in line 331. We feel the way the word is used in the abstract and in line 331 is appropriate. We say in the abstract: "The novel approach is referred to as the 'scaled-up DNS'. The scaling-up is done in two parts, first by increasing both the computational domain and the Kolmogorov microscale, and second by using super-droplets instead of real droplets". However, we believe (based on the comments below) that the reviewer feels the statement in line 133 ("and provides the key novelty for the spectral homogeneous isotropic turbulence simulation") is not appropriate.

Revision to the text: In response to the comment, we removed the part of the sentence "and provides the key novelty…" in line 331.

> Since this is a DNS study, let us put the LES aside. Eq.4-9 in the manuscript show the standard scaling argument of the scale separation of turbulence. DNS (finite difference numerical method or spectral method) of turbulence is limited by Re. To study how large scales of turbulence affect supersaturation fluctuations by means of DNS, the authors increased both the integral length scale and Kolmogorov length scale of turbulence simultaneously so that Re can be kept unchanged. This is very well described by Eq.4-9. What is the novelty regarding numerical methodology in DNS simulations with this configuration? In other words, how can I see the novelty from Eq.4-9?

Agreed, see above. Perhaps the novelty is that such a technique has never been used before no matter how elementary it is. But we agree, "novelty" is a poor choice for Eqs. 4-9. We removed this part of the sentence.

Revision to the text: the part of the sentence in question has been removed, see the previous comment.

The Reynolds number is defined as Re=u_rmsL/\nu. One can simply increase L and \nu at the same time in a DNS simulation to check how large eddies affect supersaturation fluctuations.

Yes, this is exactly what we do. But we feel the scaling to increase \nu has never been shown before. We agree, it is simple, but we have not seen it in the literature. Also, please note that we show how this can be done keeping the Reynolds number the same (Eq. 8) and then when the Reynolds number is changed (Eq. 9).

No change to the text.

I don't understand the comment about the numerical dissipation for finite difference method and the spectral method. Both methods deal with the physical dissipation, i.e., energy dissipation in the dissipation range of turbulence. How exactly increasing L and \nu simultaneously in a spectral DNS code makes the simulation novel?

This is perhaps a minor point, not worth discussing in the manuscript. The key difference is that the spectral method is by design numerically inviscid. As a result, the code blows if there is no dissipation because of the energy accumulation at the smallest scales. This is in contrast to a finite-difference code that can be run without any explicit dissipation because applying a monotone advection scheme provides sufficient small-scale dissipation. This is the idea of implicit LES that we briefly mention. For the last sentence of the comment, we have never seen DNS simulation with increased L and \nu, especially when applied to moist processes.

Revision to the text: the part of the sentence with the word "novel" has been removed (see above).

To my knowledge, statistical convergence tests of the superdroplet method in such large simulation domain of DNS (large air viscosity though) has never been done. This is new and useful for the study of diffusional growth of cloud droplets.

Yes, we agree.

2.  The authors responded that "Yes, the eddy dissipation rate is a small-scale quantity, but this is how intensity of turbulence is expressed in models and in observations".

Indeed, in the models and observations, \epsilon has been used to characterize the intensity of turbulence. However, I disagree that turbulence intensity is determined by \epsilon.

My understanding of the present study is that DNS combined with superdroplet approach is used to study the diffusional growth of cloud droplets. DNS means one solve the Navier-Stokes equation to the native scale of turbulence. Even though Re is small in DNS studies and scale contamination is inevitable, intermittency still exists, i.e., the inhomogeneous distribution of energy dissipation rate in turbulence. This intermittency is determined by the Reynolds number. That is, the energy dissipation rate is determined by

the Reynolds number.
I encourage the authors to have a look at this seminal paper
https://doi.org/10.1103/PhysRevLett.72.336 and numerous laboratory experimental
evidence and observational evidence on this topic.

Yes, we agree with these comments. First, we agree that turbulence intermittency in high Re
turbulence is important and needs to be considered. However, given the limited Reynolds
number (in either true DNS or scaled-up DNS), its impact cannot be meaningfully addressed in
our study. Second, we appreciate the fact that TKE dissipation is an intermittent small-scale
quantity that fluctuates strongly in space and time. However, the *mean* TKE dissipation is a
parameter to setup the simulations, in the same way as typically done in other homogeneous
isotropic turbulence DNS studies. Lower panels in Fig. 2 show that the mean dissipation rate
derived from the simulated enstrophy is as expected in scaled-up simulations.

We appreciate the reference.

Revision: For clarification, we replaced the statement "using the simulated enstrophy" in line
164 with "as twice the product of the scaled-up viscosity and the mean simulated enstrophy".

**Referee #2**

We respond to the comments in green.

Specific Comments

1. line 65: It is possible to do simulations with larger domains with the EMPM. It would
appropriate to mention here that the EMPM simulations reported in Su et al. (1998) used a 20-m
domain size, and EMPM domains up to 100-m domains were used in To˙lle and Krueger (2014).

To˙lle, M. H., and S. K. Krueger, 2014: Effects of entrainment and mixing on the droplet size
distributions in warm cumulus clouds. J. Adv. Model. Earth Syst., 6, 281–299,
doi:10.1002/2012MS000209

As stated in our response to 1 and 2, we do not want to bring entrainment/mixing in this
manuscript except in a brief comment in the final paragraph in the conclusion section. No
changes to the text.

lines 62-65 (revised): The authors write "From the point of view of realistic cloud modelling,
developing and validating robust subgrid-scale schemes for contemporary large eddy simulation
(LES) models (i.e., featuring grid lengths of a few tens of meters) requires performing DNS-like
simulations in computational domains comparable to the size of the LES grid box." The EMPM
does exactly this, as noted in my original comment. This capability is not limited to entraining
parcels. It seems that it would be appropriate to mention the EMPM approach as well. It is
clearly relevant to the authors' text.

We insist that our study concerns isotropic homogeneous turbulence as a model for cloud turbulence and its impact on droplet spectrum, and not entrainment and mixing. Please see the title of the paper the reviewer asks us to cite. Besides just a mention of entrainment/mixing in the introduction, we do not want to bring it in the manuscript. We simply removed the sentence staring with "From the point of view…" from the revised manuscript.

Changes to the manuscript: we removed the sentence in question.

2. line 100, Eq. (1): In general, this equation should include a term $w'd\bar{T}/dz$. I suspect that this term is missing because $\bar{T}/dz = 0$ is enforced due to the cyclic b.c. at the top and bottom boundaries. It this is the case, it should be mentioned. It should also be mentioned that forcing $\bar{T}/dz = 0$ is equivalent to forcing a non-zero gradient of potential temperature, which acts as the source of temperature and supersaturation fluctuations.

The reviewer is correct. DNS by design cannot feature mean temperature gradients because of the triply-periodic boundary conditions. This is why Eq. (1) does not have the w dT/dz term. Eq. (1) is standard for the DNS of homogeneous isotropic turbulence (e.g., see Eq. 9 in Vaillancourt et al. JAS 2001). We prefer not to bring this aspect in the model description. No changes to the text.

The source of the supersaturation fluctuations is vertical velocity fluctuations and condensation. Air parcels ascend or descend along saturated adiabats to a good approximation, so that $dT/dz = \Gamma_S$, which produces temperature fluctuations $\Delta T \approx -\Gamma_S \Delta z$ when $d\bar{T}/dz = 0$. Therefore, the specification of $d\bar{T}/dz = 0$ is important and should be mentioned.

It also is not true that "DNS by design cannot feature mean temperature gradients". If the thermodynamic variable used in the DNS is potential temperature, for example, then $d\theta/dz = 0$ would be required but $dT/dz = g/c_p$.

We do not agree. Please check published papers that apply DNS to the problem of droplet growth in *homogeneous isotropic* turbulence. We cite those papers and our study builds on them. In its dry form, vertical velocity has no impact on the temperature. In fact, the temperature equation is never solved in the dry homogeneous isotropic turbulence DNS. The extension to moist processes starting with Vaillancourt et al. includes -w g/cp term representing dry adiabatic temperature change that affects supersaturation and mimicks the impact of stratification. In fact, some solve for water vapor mixing ratio and temperature (like Vaillancourt et al.), some solve for the supersaturation alone (like Lanotte et al.). The supersaturation equation comes from combining temperature and moisture equations and neglecting small difference in molecular transport coefficients. We strongly insist that bringing this discussion to the paper would only lead to the confusion and detract from the proper context of our study.

No change to the text.

3. lines 365-6: This might be too general of a statement. The large eddies dominate for this mode of supersaturation fluctuation because they span a larger potential temperature difference for the

same mean vertical gradient. For other modes of supersaturation fluctuation generation such as entrainment, large eddies also dominate, but for a different reason (their greater mixing time scale).

This comment is incorrect. Larger eddies feature larger and longer-lasting vertical velocity fluctuations because of the way TKE scales with L for the same eddy dissipation rate. As explained above, spectral DNS has no mean vertical gradients. No changes to the text

As noted in the previous comment, it is incorrect to state the "spectral DNS has no mean vertical gradients". When $dT/dz = 0$, $d\theta/dz = \Gamma_d$, for example.

We apologize, the sentence in blue from our previous response should read "spectral DNS of *homogeneous isotropic* turbulence has no mean vertical gradients". We think the reviewer agrees.

No change to the text.

Addtional Specific Comments

4.  lines 376-383 (revised version): The authors "consider supersaturation fluctuations in a simple stochastic model of a droplet ensemble" and note that "The key advantage of the stochastic model is that its computational cost is just a tiny fraction of a DNS simulation." Furthermore, they write that "the stochastic model provides a simple and physically appealing approach to multiscale large-eddy simulation of a cloud applying Lagrangian particle-based microphysics."

    The same could be said about the EMPM (Su et al. 1998; Tölle, M. H., and S. K. Krueger, 2014) and the L3 model (Hoffmann and Feingold, 2019; Hoffmann et al., 2019). It may benefit the readers to mention these relevant studies.

    Hoffmann, F. and G. Feingold, 2019: Entrainment and Mixing in Stratocumulus: Effects of a New Explicit Subgrid-Scale Scheme for Large-Eddy Simulations with Particle-Based Microphysics. J. Atmos. Sci., 76, 1955-1973, https://doi.org/10.1175/JAS-D-18-0318.1

    Hoffmann, F., T. Yamaguchi, and G. Feingold, 2019: Inhomogeneous Mixing in Lagrangian Cloud Models: Effects on the Production of Precipitation Embryos. J. Atmos. Sci., 76, 113-133, https://doi.org/10.1175/JAS-D-18-0087.1

Again, we do not agree, sorry. Stochastic model of Grabowski and Abade (JAS 2017) as used in the paper under review is much simpler than the Liner Eddy Model used in the papers listed above. We do not feel referring to those papers is needed in the context of our manuscript.

No change in response to this comment.

5. lines 384-6 (revised version): The authors write that "The next step can be to apply this approach in a rising parcel simulations..." This statement should be qualified because in the approach described, the supersaturation fluctuations are generated by turbulent vertical motions acting on a specified and unrealistic mean gradient of temperature (isothermal rather than saturated adiabatic). See comment 2.

We strongly disagree with this comment. See our response to 2.

Change to the manuscript: We added the following sentence to follow the one listed by the reviewer: "
[revised manuscript text omitted]